# Troposphere – stratosphere integrated BrO profile retrieval over the central Pacific Ocean

Theodore K. Koenig[1,2,a], François Hendrick[3], Douglas Kinnison[4], Christopher F. Lee[1,2], Michel Van Roozendael[3], and Rainer Volkamer[1,2,*]

[1]Department of Chemistry, University of Colorado Boulder, Boulder, CO, 80309-0215, USA
[2]Cooperative Institute for Research in Environmental Sciences (CIRES), University of Colorado Boulder, Boulder, CO, 80309-0216, USA
[3]Royal Belgian Institute for Space Aeronomy (BIRA-IASB), Brussels, 1180, Belgium
[4]Atmospheric Chemistry Observations & Modeling (ACOM), National Center for Atmospheric Research (NCAR), Boulder, CO 80301, USA
[a]Now at: State Key Joint Laboratory of Environmental Simulation and Pollution Control, BIC-ESAT and IJRC, College of Environmental Sciences and Engineering, Peking University, Beijing 100871, China

*Correspondence to*: Theodore K. Koenig (theodore.k.koenig@pku.edu.cn) and Rainer Volkamer (rainer.volkamer@colorado.edu)

**Abstract.** Bromine monoxide (BrO) is relevant to atmospheric oxidative capacity, affects the lifetime of greenhouse gases (i.e., methane, dimethylsulfide), and mercury oxidation. However, measurements of BrO radical vertical profiles are rare, and BrO is highly variable. As a result, the few available aircraft observations in different regions of the atmosphere are not easily reconciled. Autonomous MAX-DOAS placed at remote mountaintop observatories (MT-DOAS) presents a cost-effective alternative to aircraft with potential to probe the climate-relevant yet understudied free troposphere more routinely.

Here we describe an innovative full-atmosphere BrO and formaldehyde (HCHO) profile retrieval algorithm using MT-DOAS measurements at Mauna Loa Observatory (19.536°N; 155.577°W; 3401m asl). The retrieval is based on time-dependent optimal estimation, and simultaneously inverts 190+ individual BrO (and formaldehyde, HCHO) SCDs (slant column densities, $SCD = dSCD + SCD_{Ref}$) from solar stray light spectra measured in the zenith and off-axis geometries at high and low solar zenith angle (92° > SZA > 30°) to derive BrO concentration profiles from 1.9 to 35 km with 7.5 degrees of freedom (DoF) . Two case study days are characterized by the absence (26 Apr 2017, base case) and presence of a Rossby Wave breaking double tropopause (29 Apr 2017, RW-DT case). Stratospheric BrO vertical columns are nearly identical on both days (VCD = $(1.5 \pm 0.2) \times 10^{13}$ molec cm$^{-2}$), and the stratospheric BrO profile peaks at a lower altitude during the RW-DT (1.6 – 2.0 DoFs). Tropospheric BrO VCDs increase from $(0.70 \pm 0.14) \times 10^{13}$ molec cm$^{-2}$ (base case) to $(1.00 \pm 0.14) \times 10^{13}$ molec cm$^{-2}$ (RW-DT), owing to a three-fold increase in BrO in the upper troposphere (1.7 – 1.9  DoF). BrO at MLO increases from $(0.23 \pm 0.03)$ pptv (base case) to $(0.46 \pm 0.03)$ pptv BrO (RW-DT), and is characterized with added time resolution (~3.8 DoF). Up to $(0.9 \pm 0.1)$ pptv BrO is observed above MLO in the lower free troposphere in absence of the double tropopause.

We validate the retrieval using aircraft BrO profiles and in-situ HCHO measurements aboard the NSF/NCAR GV aircraft above MLO (11 Jan 2014) that establish BrO peaks around 2.4 pptv above 13 km in the UTLS during a similar RW-DT event ($0.83 \times 10^{13}$ molec cm$^{-2}$ tropospheric BrO VCD above 2 km). The tropospheric BrO profile measured from MT-DOAS

(RW-DT case) and the aircraft agree well (after averaging kernel smoothing). Furthermore, these tropospheric BrO profiles over the Central Pacific are found to closely resemble those over the Eastern Pacific Ocean (2-14 km); and contrast with the Western Pacific Ocean, where a C-shaped tropospheric BrO profile shape had been observed.

## 1 Introduction

Bromine has been of particular interest to atmospheric chemists since its role in the stratospheric ozone ($O_3$) hole prompted the Copenhagen Amendment to the Montreal Protocol. Gas-phase atmospheric bromine can be divided into organic species and inorganic species ($Br_y$). $Br_y$ can be further divided into active bromine ($BrO_x \equiv Br + BrO$), reservoir species (HBr, HOBr, $BrNO_2$, and $BrONO_2$), and photolabile compounds ($Br_2$ and BrCl). Atomic bromine (Br) and bromine monoxide (BrO) radicals rapidly interconvert, primarily by reaction with $O_3$ and photolysis. The atmospheric budget of $BrO_x$ – and to a

large extent $Br_y$ – is principally constrained by measurements of bromine monoxide (BrO) utilizing Differential Optical Absorption Spectroscopy (DOAS).

$BrO_x$ impacts chemistry in the troposphere and stratosphere, by modifying ($O_3$) but also in other ways. First, the most direct impact on $O_3$ is direct catalytic photochemical destruction (von Glasow et al., 2004; Read et al., 2008; Saiz-Lopez and von Glasow, 2012; Simpson et al., 2015; Schmidt et al., 2016; Wang et al., 2015; Wofsy et al., 1975). Second, $BrO_x$ modifies

$NO_x$ ($\equiv NO + NO_2$) (Evans et al., 2003; Custard et al., 2015; Lary, 2005), increasing the ratio of $NO_2/NO$ (Bloss et al., 2010), and as an overall sink of $NO_x$ is suppressing $O_3$ production (Schmidt et al., 2016). Third, $BrO_x$ increases oxidative capacity and exerts a number of competing effects on $HO_x$ ($\equiv OH + HO_2$) (Stone et al., 2018). Fourth, bromine atoms are understood to be the primary oxidant of mercury in the atmosphere (Holmes et al., 2006; Coburn et al., 2016; Goodsite et al., 2004; Shah et al., 2021) and an important loss mechanism for dimethyl sulfide (Boucher et al., 2003). Fifth, the net-effect of $BrO_x$

impacts on $O_3$, $NO_x$, and $HO_x$ leads to an increase of the lifetime of CO, hydrocarbons, and climate-active gases such as methane (Lelieveld et al., 1998; Parrella et al., 2012; Saiz-Lopez and von Glasow, 2012; Sherwen et al., 2016; Wang et al., 2021).

Remote sensing of BrO by DOAS from the ground has previously been conducted near the poles (Kreher et al., 1997; Sinnhuber et al., 2002; Schofield et al., 2006; Hendrick et al., 2007), at mid-latitudes (Aliwell et al., 1997, 2002; Sinnhuber

et al., 2002; Schofield et al., 2004), in the subtropics (Leser et al., 2003; Coburn et al., 2011, 2016), and in the tropics (Theys et al., 2007). There is also an extensive record of DOAS measurements of BrO from space (Chance, 1998; Wagner et al., 2001; Richter et al., 2002; Hendrick et al., 2009; Theys et al., 2011; Seo et al., 2019). DOAS measurements of BrO in free troposphere and lower stratosphere from balloons (Fitzenberger et al., 2000; Pundt et al., 2002; Dorf et al., 2006, 2008) and from aircraft (Volkamer et al., 2015; Wang et al., 2015; Werner et al., 2017; Koenig et al., 2017) find that BrO is widespread

in the free troposphere but highly variable. Over the Pacific, previous tropospheric measurements have found that BrO mixing ratios increase roughly linearly with altitude over the eastern Pacific being near or below detection in the boundary layer and greatest below the tropopause (Volkamer et al., 2015; Wang et al., 2015; Dix et al., 2016), while measurements by

DOAS (Koenig et al., 2017) and other methods (Le Breton et al., 2017; Chen et al., 2016) find a more C-shaped profile over the western Pacific. The sensitivity of ground-based and space-based measurements to BrO in the free troposphere was apparent even prior to the first profile studies (Harder et al., 1998; Frieß et al., 1999; Van Roozendael et al., 2002). However, the quantification and location of BrO in the free troposphere from the ground and space requires accurate knowledge of the air mass factors (AMF), which the variability in tropospheric profiles continues to confound.

This work systematically explores the significant information content of a MountainTop (MT)-DOAS to profile tropospheric and stratospheric trace gases, which has potential to help overcome this limitation. Figure 1 illustrates the measurement concept: 1) ZS-DOAS measurements under twilight conditions exploit the motion of the sun varying atmospheric path length and scattering attitude to profile the stratosphere, while 2) Off-AXis (OA)-DOAS measurements during the day profile the troposphere. Section 2 introduces the hardware and software methods used; including the DOAS instrumentation (sect. 2.1), DOAS fit settings (sect. 2.2), radiative transfer models used to account for repartitioning of bromine during twilight (sect. 2.3); data taken from global models (sect. 2.4), and introduces the time-dependent Optimal Estimation approach (accounts for non-photochemical diurnal variability, sect. 2.5). Section 3 introduces the case studies (sect. 3.1) and discusses the results; including sensitivity studies in DOAS fitting that exploit recent advances in the knowledge of absorption cross-section spectra (sect. 3.2), the full atmosphere profile retrievals (sect. 3.3), and evaluation of the profiles using aircraft measurements (sect. 3.4); finally, the results are placed in context with previous aircraft campaigns (sect. 3.5). Section 4 presents conclusions and gives an outlook on opportunities for future work.

## 2 Instrumentation and Methods

Data reported herein were principally collected by two Multi-AXis (MAX)-DOAS instruments: principally a University of Colorado (CU) mountaintop MT-DOAS instrument at Mauna Loa Observatory (MLO) described in 2.1.1. References to the MT-DOAS hereafter refer to this instrument unless otherwise specified; and the CU Airborne MAX-DOAS (AMAX-DOAS) described in 2.1.2 collected during the CONvective TRansport of Active Species in the Tropics (CONTRAST) field campaign.

### 2.1 DOAS Instrumentation

The MT-DOAS and AMAX-DOAS are broadly similar; solar scattered light enters a telescope and is transported by fiber optic cables to two diffraction spectrographs, which image and save spectra for analysis. The instruments are described below.

### 2.1.1 MLO MT-DOAS

The CU MT-DOAS at MLO (19.536°N 155.577°W 3401m a.s.l.) is a modified version of that previously deployed at Pensacola, Florida (Coburn et al., 2011). It is also a near-copy of an instrument which participated in the Cabauw Intercomparison of Nitrogen Dioxide Measuring Instruments-2 (CINDI-2) campaign (Kreher et al., 2020) matching apart

from some precise points of the form factor and spectroscope alignment. The system was deployed to MLO in February 2017 and has operated near-continuously since then. The system will be described briefly with modifications since the Pensacola deployment and particulars of the setup at MLO highlighted with further details in the supplement.

The telescope gathers light from a symmetric cone of $0.3°^2$. The anodized aluminum telescope has been ruggedized for the environment on MLO. The ¼" baseplate has been replaced with 1" of cast aluminum to mitigate possible warping from extended mechanical stress from mounting during long term deployments and to provide additional mass as a defense against winds. In addition, a heating element was placed inside the telescope head to provide heating when the temperature dropped below 0°C as indicated by a sensor also in the housing.

The primary azimuthal viewing direction of the telescope is estimated to be along -50±2°. The telescope head rotates in a plane of elevation angles (EA) defined relative to horizontal (positive up, negative down). The telescope has a free line of sight for EA >-4.5° in the primary direction, below which major radiative effects from the reflective roof of the Chin Building are always apparent, and EA >7° in the reverse direction (heading 130±2°), below which the line of sight intercepts the ridge of Mauna Loa.

The instrument includes two spectrometers: an Acton SP2356i spectrometer with a PIXIS 400B CCD detector equipped with UV fluorescence coating (AP) covering 307.6 - 474.8 nm and an Ocean Optics QE65000 spectrometer (QE) covering 414.3 - 1119.7 nm. Atomic emission spectra from Hg and Kr are used to determine the spectral resolutions of 0.593 nm per full-width at half-maximum (FWHM) and 1.45 nm for the AP and QE respectively. Spectra collected on the AP can be analyzed for BrO, HCHO, $NO_2$ (360 nm and 450 nm), $O_2$-$O_2$ (360 nm; hereafter $O_4$), glyoxal (CHOCHO), and iodine oxide (IO) radicals. Spectra on the QE can be analyzed for $NO_2$ (450 nm and 560 nm), $O_4$ (477 nm), CHOCHO, and $H_2O$. In this work only BrO, HCHO, $NO_2$, and $O_4$ are presented.

All spectra from the MT-DOAS used in the analysis are collected with one minute total integration time. The spectrometers are operated synchronously. For 75° < SZA < 110° spectra were collected in a ZS geometry. For SZA > 92.5° insufficient photons are collected for meaningful analysis on the AP and the data are excluded. For SZA<75° spectra were collected in the following standard sequence of angles where angles preceded by † are collected in the reverse azimuthal direction (+130°±2): -4.5°, -3°, -2°, -1°, 0° (four repeats), 1°, 2°, 3°, 5°, 8°, 12°, 20°, 30° (four repeats), 45°, 90° (four repeats), †45°, †30° (four repeats), †20°, †12°, †8°. Spectra in both directions were analyzed, however, it was subsequently found that the two viewing directions could not be reproduced simultaneously with 1D radiative transfer modelling and the data from the reverse direction are not reported here.

For fixed reference analyses, spectra for both days were analyzed against a four-minute zenith acquisition collected shortly before local noon on April 26 (Apr 26 21:18 UTC, 11:18 local, EA = 90°, SZA = 15.92°), which is later in the morning than the data presented in this work. For moving reference analyses, the fixed reference analyses are adjusted by the fitted zenith spectra linearly interpolated in time which was found to obtain results not statistically distinguishable from irradiance interpolation but is much more time efficient.

## 2.1.2 AMAX-DOAS

The CU AMAX-DOAS instrument has been deployed during field campaigns in urban air (Oetjen et al., 2013; Baidar et al.,
2013), and over remote oceans (Dix et al., 2013; Volkamer et al., 2015), and is described in detail in these papers. The
configuration employed during the CONTRAST field campaign is described in (Koenig et al., 2017).

Briefly, the CU AMAX-DOAS consists of a wing-mounted pylon containing a motion-stabilized telescope, and two
spectrographs housed in the interior of the aircraft. One Acton SP2150/PIXIS400B CCD unit (AP1) covers the spectral range
from 330-470 nm with 0.7 nm full width half maximum (FWHM) optical resolution based on the 404.66 nm Hg line. The
140 other (AP2) measures 440-700 nm at 1.2 nm FWHM resolution based on the 450.24 Kr line. Spectra collected on AP1 are
used for the measurement of BrO, glyoxal, HCHO, $H_2O$, IO, $NO_2$ (360 nm and 450 nm), and $O_4$ (360 nm); spectra on AP2
are used for the measurement of $NO_2$ (560 nm) and $O_4$ (477 nm). In this work only BrO and $O_4$ (360 nm) results will be
presented, using EA 0° spectra collected with 30 s integration times.

The data presented here were collected towards the end of research flight #1 (RF01; January 11, 2014) of the CONTRAST
field campaign, when the aircraft conducted a deep vertical profile in the vicinity of MLO. For RF01 spectra were analyzed
against a two-minute zenith acquisition collected during the period presented (Jan 12 00:47 UTC, Jan 11 14:47 local,
155.81°W, 20.12°N, 3.02 km a.s.l., EA = 90°, SZA = 53.28 °). Flight segments are designated following a system described
more fully in Koenig et al., (2017), in brief  monotonic ascents and descents for a given flight are assigned sequentially as
(RF##-aa) such that all ascents have odd numbers for aa and all descents have even-numbered aa.

## 2.2 DOAS fitting

Trace gases were fit using the DOAS method (Platt and Stutz, 2008) using the QDOAS software package (Danckaert et al.,
2012). Fit settings for the MT-DOAS are summarized in Table S1 and for the AMAX-DOAS in Table S2. The wavelength
calibration for each spectrum is precisely determined by measuring the atomic emissions lines as described in Section 2.1.
This slit function was further refined by fitting two wavelength dependent width parameters, using the Kurucz spectrum as
reference (Chance and Kurucz, 2010; Kurucz et al., 1984). The slit function was fixed for final calibration of the instrument
wavelength mapping. High resolution laboratory cross-sections (species and references given in Table S1) are convolved
with the instrument function for analysis. Broadband extinction including Mie and Rayleigh scattering is accounted for by a
polynomial. Trace gases with broad band absorption components such as $O_3$ and $NO_2$ are orthogonalized to this polynomial.
A linear intensity offset is included to account for instrumental stray light and imperfect knowledge of the Grainger-Ring
effect (Grainger and Ring, 1962). Absorption by relevant species is fitted simultaneously using the non-linear Marquardt-
Levenberg algorithm with full non-linear treatment reserved for shift, stretch, and intensity offset (Danckaert et al., 2012).
This is done in finite wavelength windows targeting specific trace gases. The determination of optimized fit settings is a

major product of this work, therefore relevant sensitivity studies and final fit settings are discussed in greater length Sect. 3.2 and 3.3 and presented in Fig. 3.

## 2.3 Radiative Transfer

Two radiative transfer codes were used for this study. For ZS-DOAS measurements, Discrete Ordinate Method Radiative Transfer (DISORT) was used and for OA measurements the Monte Carlo Atmospheric Radiative Transfer Inversion Model (McArtim) was used. Weighting functions used for the retrieval were calculated from both models using the same vertical grid, consisting of the following layers (given as altitude a.s.l.): 0 - 0.9 km, 0.5 km layers between 0.9 km to 3.4 km (instrument altitude), 0.5 km layers from 3.4 km to 7.4 km, 2 km layers from 7.4 to 53.4 km. Layers below the instrument (necessary for downward-looking angles) were not represented in DISORT, while longitudinal modeling of atmospheric change along the solar light path (necessary for high SZA) was not modeled in McArtim. For SZA < 80° and altitudes above ~7.4 km the results from both models agreed within 1.06% RMS difference.

### 2.3.1 DISORT with PSCBOX

The principal forward model for stratospheric and ZS-DOAS measurements was the UVspec/DISORT package (Mayer and Kylling, 2005) which utilizes the discrete ordinate method in a pseudo-spherical geometry approximation. The application of the model to twilight measurements by ZS-DOAS is described in detail in Hendrick et al. (2004) and it has been utilized since for BrO (Hendrick et al., 2007; Theys et al., 2007, 2011).

The model was run in multiple scattering mode including Rayleigh and Mie scattering and molecular absorption. Pressure and temperature profiles were based on those from the CAM-Chem model (see Sect. 2.4). Stratospheric aerosol was modeled to represent background conditions and tropospheric aerosol was derived from inversions reproducing the observed $O_4$, in the course of which albedo was also optimized (see Supplement for details). UVspec/DISORT was run with the instrument at the surface, placed at 3.4 km altitude, with only the layers above this altitude treated in the model. Rapid photochemical changes at twilight cause concentrations, particularly of BrO and $NO_2$, to change along the light path impacting radiative transfer (Sinnhuber et al., 2002). This is represented in UVspec/DISORT by introducing a second dimension with the different profiles along the light path populated using the stacked box photochemical model PSCBOX (Errera and Fonteyn, 2001; Hendrick et al., 2004). PSCBOX was run with 20 altitude levels between ~10 and ~55 km (altitudes below 10 km were assumed to have the same chemical partitioning as the lowest level) with chemical species from the SLIMCAT 3-D chemical transport model (Chipperfield, 2006; Hendrick et al., 2007). The model has been updated to reflect the latest bromine chemistry taken from the JPL 2015 compilation (Burkholder et al., 2015).

While McArtim is in principle capable of modeling stratospheric radiative transfer at twilight, a suitable 2D implementation of McArtim to use in conjunction with a photochemical model (i.e., UVspec) was not straightforward. We use DISORT instead since the 2D geometry is defined along the solar azimuth, which is ideal for accounting for photochemical effects

along the principle line of sight. For ZS data this agrees with McArtim results including lower altitudes to better than ~1% differences.

### 2.3.2 McArtim

The principal forward model for MAX-DOAS measurements and aircraft measurements was McArtim (Deutschmann et al., 2011) in a 1D spherical atmosphere. The model includes Rayleigh and Mie scattering and molecular absorption. Pressure
and temperature profiles were based on those from the CAM-Chem model (see Sect. 2.4). Tropospheric aerosol was assumed to be marine for both MT-DOAS and AMAX-DOAS simulations: non-absorbing Henyey-Greenstein aerosol phase function with asymmetry parameter $g = 0.72$ above the boundary layer and $g = 0.77$ in the boundary layer. The retrieval of aerosol extinction was based on reproducing $O_4$ signals measured by DOAS at 360 nm (Spinei et al., 2015; Volkamer et al., 2015) and utilized a layering approach. The surface was set at sea level with an albedo of 0.05 at 360 nm and 0.08 at 477 nm. For
MT-DOAS, sensitivity studies were conducted for the surface altitude and albedo (see Supplement for details).

For MT-DOAS retrievals, the atmosphere was initialized with a 200 m grid from the surface to 7.4 km altitude. Pressure, temperature, humidity and major absorber ($O_3$ and $NO_2$) profiles were based on those from the CAM-Chem model (see Sect. 2.4). Aerosol conditions for the data in this work had significant extinction below the instrument but sub-Rayleigh extinction near and above the instrument (see Fig. S5, Sect. 3.3.1). For these conditions, each EA was given an altitude sensitivity
mapping and the extinction profile adjusted starting from lower EA and lower altitudes for better agreement with $O_4$ SCDs. The a priori aerosol profile used consisted of constant aerosol extinction below 2 km then exponentially decreasing with altitude with a scale height of 2 km, with the magnitude first manually determined for approximate agreement. EA were proceeded through from lowest (-4.5°) to highest (45°) This bottom-up as opposed to top-down approach was chosen because representing aerosols/clouds below instrument altitude was needed to reproduce $O_4$ observations at higher EA. The
bottom-up sequence of $O_4$ comparisons and adjustments was run six times independently for each scan. For morning twilight measurements, the aerosol profile from the first scan was used.

The aerosol extinction used for AMAX-DOAS retrievals is based on that used in (Baidar et al., 2013). In brief, aircraft limb measurements are collected on a 200 m grid with aerosol added to or removed from layers until the average difference between simulated and measured $O_4$ signals is within a specified tolerance. This procedure is repeated for each grid layer
from the top of the profile down then iterated with decreasing tolerance. Clouds which are present are introduced based on camera imagery from the GV, assumed to have a constant optical density distribution the magnitude of which is adjusted manually to reproduce signals measured above and below the cloud. Pressure, temperature, humidity and major absorber ($O_3$ and $NO_2$) profiles were based on aircraft measurements below the aircraft ceiling altitude and based on CAM-Chem above this altitude.

## 2.4 CAM-Chem

The 3-D chemistry climate model Community Atmospheric Model with Chemistry (CAM-Chem) is described in Lamarque et al. (2012). In the present configuration, meteorological analysis (from GEOS-5) specific to the observational periods are used to constrain the meteorological fields (horizontal wind components and temperature) in CAM-Chem. The horizontal resolution is $0.9° \times 1.25°$ and vertical resolution of 52 levels includes full coverage of the troposphere and stratosphere, with a full representation of tropospheric and stratospheric chemistry.

For bromine chemistry, the version used here includes geographically-distributed and time-dependent oceanic emissions of six bromocarbons ($CHBr_3$, $CH_2Br_2$, $CH_2BrCl$, $CHBrCl_2$, $CHBr_2Cl$, $CH_2IBr$) as well as an additional source of inorganic bromine and chlorine in the lower troposphere due to sea-salt aerosol (SSA) dehalogenation (Ordóñez et al., 2012; Saiz-Lopez et al., 2012; Fernandez et al., 2014). It considers heterogeneous processes for halogen species on a variety of surfaces including uptake and recycling of HBr, HOBr, and $BrONO_2$ on ice-crystals and sulfate aerosols.

CAM-Chem fields for pressure, temperature, water vapor, $O_3$, and $NO_2$ were used to constrain the atmosphere for MT-DOAS radiative transfer simulations, and above flight altitude for AMAX-DOAS radiative transfer simulations. For MT-DOAS measurements data was interpolated from the surrounding points horizontally then averaged from the 30-minute output over the period of the included scans (the $NO_2$ field was still adjusted in spatiotemporally based on PSCBOX see Sect. 2.3.1). For AMAX-DOAS, data was averaged from extracted curtains for the individual profiles. CAM-Chem profiles extracted for the purposes of comparison with measurements were extracted the same way.

## 2.5 Retrieval Methods

The BrO profiles reported herein were retrieved using optimal estimation (Rodgers, 2000). In brief, the MT-DOAS measurement vector, $\mathbf{y}$, consists of SCDs (= dSCD + $SCD_{Ref}$, Sect. 2.2 and 2.5.1.), where each measurement is a function of the absorber profile $\mathbf{x}$ and weighing functions $\mathbf{K}$ (determined by RTM, Sect. 2.3) such that $\mathbf{y} = \mathbf{K}\,\mathbf{x}$. The solution to the inverse problem consists in finding the maximum likelihood estimator of the profile $\hat{\mathbf{x}}$, given an a priori profile $\mathbf{x_a}$, the measurements $\mathbf{y}$, and their respective covariance matrices $\mathbf{S_a}$ and $\mathbf{S_\varepsilon}$. Because BrO is an optically thin absorber i.e. $\mathbf{K}$ is independent of $\mathbf{x}$, and for this linear case the solution is given by:

$$\hat{\mathbf{x}} = \mathbf{x_a} + \mathbf{S_a}\,\mathbf{K}^T\,(\mathbf{K}\,\mathbf{S_a}\,\mathbf{K}^T + \mathbf{S_\varepsilon})^{-1}\,(\mathbf{y} - \mathbf{K}\,\mathbf{x_a}) \qquad (1)$$

The MT-DOAS retrieval was developed for this work and includes a number of steps described in more detail below. The AMAX-DOAS optimal estimation inversion was more straightforward. HCHO and BrO dSCDs were used with the DOAS fit errors employed for the uncertainties. The three individual aircraft profiles (two descents and one ascent; Sect. 3.4) were inverted separately.

### 2.5.1 Photochemical Langley Plot

For the MT-DOAS inversion, SCDs were used as the elements of $\mathbf{y}$, which required the determination of $SCD_{Ref}$. The primary method of doing so was through use of a Langley plot of dSCDs against air mass factors (AMF). However, given that the BrO profile changes rapidly during twilight both as a function of SZA and along the light path, a traditional Langley plot analysis is not possible. We employ a method adapted from Hendrick et al. (2007) which normalizes the AMF to a single reference SZA (70°) correcting for photochemical effects. Specifically we define $AMF_{70°SZA}(SZA) = SCD_{modeled}(SZA)$ / $VCD(70°)$, where $SCD_{modeled}(SZA)$ is computed using DISORT and PSCBOX as described above in Sect. 2.3.1. Particular instances of DISORT PSCBOX are selected based on the month of year, latitude, and bromine loading and interpolated to more precisely match a preliminarily chosen value of $VCD(70°)$. This allows for an adapted Langley equation to be defined, namely:

$$dSCD (SZA) = VCD(70°) \; AMF_{70°SZA}(SZA) - SCD_{Ref} \qquad (2)$$

Where the intercept of a fitted line determines $SCD_{Ref}$, and the slope determines $VCD(70°)$. Significant deviations from linearity would indicate that the time evolution of the BrO profile is not that assumed in the determination of $SCD_{modeled}(SZA)$, and thereby in $AMF_{70°SZA}(SZA)$. The value for $SCD_{Ref}$ determined by the Langley plot is compared and assessed using the final retrieval results (see Sect. 3.3 for details). $SCD_{Ref}$ is added to the dSCDs to obtain SCDs for optimal estimation.

### 2.5.2 Integrated Optimal Estimation

The integrated MT-DOAS retrieval required using weighting functions from DISORT and McArtim in tandem. As noted above, to accomplish this DISORT was used for weighting functions for SZA > 70° (with McArtim used to extend these below 3.4 km) and McArtim was used for the weighting functions for SZA < 70°. Both sets of weighting functions were adjusted to account for the photochemical change in a manner similar to that for AMFs described in Sect. 2.5.1. $\mathbf{K}$ is elementwise multiplied by ($\mathbf{x_{modSZA}}/\mathbf{x_{mod70°SZA}}$), where $\mathbf{x_{modSZA}}$ is the BrO profile at some SZA and $\mathbf{x_{mod70°SZA}}$ is the profile at SZA=70°. Note, because profile shape changes as a function of SZA this adjustment varies with altitude unlike that described for AMF.

For each day, all measurements were considered simultaneously under the assumption of constant $Br_y$, i.e. that the changes in the BrO profile are captured by the photochemical model and reflect only chemical repartitioning.

The MT-DOAS results reported herein used an a priori BrO profile composed of 0.5 pptv below 17.4 km, and set to 50% of the stratospheric climatology (Theys et al., 2011) above 17.4 km. For the measurement covariance $\mathbf{S_\varepsilon}$ the uncertainties were assumed to be uncorrelated and equivalent to the DOAS fit errors. Sensitivity studies were conducted accounting additionally for the uncertainty in the BrO cross-section (~10.5%) and the constrained DOAS fits. The a priori covariance $\mathbf{S_a}$, was computed as 100% of the a priori below 15.4 km and 50% above this altitude, with Gaussian correlation heights (Clémer

et al., 2010) of 1 km and 4 km below and above 15.4 km altitude, respectively. This results in the diagonal elements of $S_a$ being relatively similar with altitude compared to using the climatology.

The conventional approach to retrieving trace-gas profiles changing in time from data acquired by a MAX-DOAS instrument is to retrieve separate profiles from individual OA angle scans – often in a moving reference analysis to minimize the effect of signals mostly captured by ZS data. Where information content is limited, scans might also be averaged or otherwise combined. One limitation of such an approach is how to combine it with a stratospheric profile retrieved using ZS-DOAS data. Using the nearest zenith reference removes most dependence on the stratospheric profile but to our knowledge existing approaches effectively impose a time-dependence on the stratosphere (most typically constancy) as does our time-independent retrieval. Furthermore, the constraint although conceptually on the stratosphere also includes altitudes where the information content from an individual OA scan is limited and ZS variation contributes significantly such as the upper troposphere and the various constraints are tied together. The different scans are contingent on the imposed trend and not statistically fully independent which can obscure the statistical significance of comparing scans. Here we define an alternate approach where a consistent time function $f(t)$ for changes in the profile is used but different layers in the atmosphere can vary separately.

### 2.5.3 Time-Dependent Retrieval

The conventional time independent retrieval assumes constant $Br_y$ (a static atmosphere), with the only changes in BrO being those predicted by the photochemical model ($Br_y$ repartitions as a function of SZA). This assumption was ultimately found to be invalid for one day where dynamical changes in $Br_y$ were observed, in addition to chemical repartitioning (see Sect. 3.3 and 3.4 for details). We addressed this by augmenting the optimal estimation with time-dependent variables. To our knowledge such an approach has not been employed for DOAS optimal estimation before, so we describe it here in detail.

We define the time evolution of the BrO profile at time t ($\mathbf{x}$) in terms of L altitude regions (here L = 4) consisting of a weighted set of related atmospheric grid layers ($\mathbf{W^L}$) where $Br_y$ is expected to evolve consistently such that:

$$\mathbf{x} = \mathbf{x}_0 \left( 1 + f(t) \sum_L \mathbf{W^L C^L} \right)$$

(3)

Where $\mathbf{x_0}$ is a vector, i.e., the BrO profile at some reference time ($t_0$) with dimensions of $1 \times 36$ for this work, and $f(t) : t \rightarrow$ (-1, 1] is a scalar time evolution function such that $f(t_0) = 0$. For convenience we choose $t_0$ to match SZA = 70° so that photochemical and dynamical effects vary on a common time axis. In principle, $f(t)$ could be indexed to the layers L and folded into the sum. However, for this work there was insufficient external information to constrain more than one choice; a single $f(t)$ function describes the relative time variation in all four altitude regions. In practice, only a linear and a ramp function form – which mirrors the stratospheric $O_3$ column on Apr. 29 were tested for $f(t)$. The choice of the codomain (-1, 1] is more generally important for reasons outlined below.

$\mathbf{W}^L \rightarrow (0, 1]$ is a matrix of altitude weights defining the mapping of altitude regions L onto the altitude grid that $\mathbf{x_0}$ is defined on constructed such that $\Sigma_L \mathbf{W}^L \leq 1$ for all altitudes with dimensions of 36×4 for this work. For this work the stricter condition that $\Sigma_L \mathbf{W}^L = 1$ for all altitudes is fulfilled. Finally, $\mathbf{C}^L$ are scaling factors describing the proportional change in the BrO profile within altitude region L, with dimensions of 1×4 for this work e.g. if an element of $\mathbf{C}^L$ has a value of 1.1, BrO increased by 10% in that layer when $f(t) = 1$.

We choose to fully constrain $f(t)$ and $\mathbf{W}^L$ as such the combination of $\mathbf{x_0}$ and $\mathbf{C}^L$ fully specifies the state vector $\mathbf{x}$. We seek to retrieve $\mathbf{C}^L$ in addition to $\mathbf{x_0}$ given a choice of $f(t)$ and $\mathbf{W}^L$.

We seek to derive a time-augmented Jacobian starting from the transformation $H$ in terms of a left-side transform matrix $\mathbf{K_0}$:

$$H(\mathbf{x}) = \mathbf{K}_0 \mathbf{x} = \mathbf{K}_0 \mathbf{x}_0 \left( 1 + f(t) \sum_L \mathbf{C}^L \mathbf{W}^L \right)$$
(4)

Taking the relevant partial derivatives:

$$\frac{\partial H}{\partial x_0} = \mathbf{K}_0 \left( 1 + f(t) \sum_L \mathbf{C}^L \mathbf{W}^L \right)$$
(5a)

$$\frac{\partial H}{\partial \mathbf{C}^L} = \mathbf{K}_0 \, \mathbf{x}_0 \, f(t) \, \mathbf{W}^L$$
(5b)

A close observer may notice a potential challenge posed by these equations; the weighting functions for $\mathbf{x_0}$ require knowledge of $\mathbf{C}^L$ while the weighing functions for $\mathbf{C}^L$ require knowledge of $\mathbf{x_0}$. To resolve this challenge we take advantage of the fact that the extended definition for $\mathbf{x_0}$ still contains a term which is fully independent of $\mathbf{C}^L$ and approaches the time-independent formulation as $t \rightarrow t_0$. We therefore leverage the time-independent retrieval (Section 2.5.2; already photochemically indexed to $t_0$) to gain imperfect knowledge of $\mathbf{x_0}$. This retrieval already averages over the time dependence and should get us close to the true state as such we supply it as the a priori for $\mathbf{x_0}$ to compute the weighting functions. If the solution is too far from this a priori however, the computed partial derivatives might no longer be locally valid. To limit this effect we reduce the a priori covariance for the spatial variables ($\mathbf{x_0}$) by a factor of ten. In reporting results we use the spatial variables ($\mathbf{x_0}$) retrieved using the procedure in Section 2.5.2 including their corresponding AVK. The values and AVK for the time dependence vector ($\mathbf{C}^L$) are reported for the results of this second stage.

It remains to choose $f(t)$ and $\mathbf{W}^L$. Examination of Eq. 5a reveals the rationale for setting the codomain of $f(t)$ to (-1, 1], for solutions in which the entire retrieved column entirely disappears or doubles (considered reasonable bounding cases) this bounds the time dependent term to (-L, L] which for the small values of (L = 4) considered here is comparable to the time independent weight fixed at 1, hence ensuring the relative importance of measurements is at least partly preserved. In addition, by defining all $f(t)$ to have a maximum value of 1, the profile which is maximally different from that at $t_0$ is readily

computed and compared. For this work we considered only linear and ramp functions, ultimately using a ramp function defined to be zero prior to 70° SZA and increasing to 1 which matches an observed trend in $O_3$ VCDs (see Supplement for details). For $\mathbf{W}^L$ logistic curves were chosen as the functional form with a logistic steepness in all cases of 2 km$^{-1}$. The atmosphere was first divided at the tropopause at 17.5 km. Then at 6 km and 10 km based on the results of retrievals using single scans and modeled behavior in CAM-chem (see Sect. 3.3 for details).

The solutions to the inversion of the time-dependent retrieval were found to be highly sensitive to the a priori supplied for $\mathbf{C}^L$ including non-physical results. We suspect that this might be because the assumptions for the staged retrieval only work where the partial derivatives are sufficiently flat, or perhaps at least smooth. This necessitated systematic sensitivity studies to find solutions which were physical as well as categorizing solutions based on minimizing any time-dependent trend in the a posteriori residuals. This methodology identified a family of solutions with similar values that met stringent criteria, from which the solution with lowest overall residual was selected (see Sect. 3.3 for details).

## 3 Results and Discussion

### 3.1 Rossby Wave Breaking as a Natural Experiment

As outlined in the introduction there is a well-established record of measurements of BrO in the stratosphere by ZS-DOAS and near instrument altitude by OA-DOAS. The sensitivity of an integrated retrieval to the region of limited information content in the upper troposphere can be tested when a change in BrO is expected in this region of the atmosphere. The much larger mixing ratios in the stratosphere (>5 pptv BrO) compared to those in the troposphere (0.5 – 1.0 pptv BrO) can result in high concentrations during Rossby wave breaking double tropopause (RW-DT). Maua Loa is seasonally located near the subtropical jet where the tropopause — and associated BrO gradient — dramatically change altitude from ~17 km on the equatorward side to ~12 km on the poleward side. RW-DT events can transport BrO from the mid-latitude stratosphere into the tropical upper troposphere along isentropes. Such events are particularly frequent in the vicinity of the Hawaiian islands (Wernli and Sprenger, 2007; Funatsu and Waugh, 2008).

Routine and rapid processing of ZS-DOAS data for $O_3$ VCDs can readily identify RW-DTs as $O_3$ VCD enhancements. However, RW-DTs are found to be shortly preceded by large-scale convection, and indeed might have a causal link to it (Funatsu and Waugh, 2008; de Vries, 2021). This convection in turn leads to cloud conditions which challenge or preclude DOAS retrievals. Over the course of five years of observations only a limited number of unambiguous RW-DT have been observed (Fig. S1) and 29 April 2017 was an unusual instance when a RW-DT moved over MLO in such a manner as to allow significant periods of both ZS- and OA-DOAS which was chosen for this study.

Fig. S2 highlights the natural experiment that the RW-DT provides as modeled by CAM-chem. Regional conditions on the morning of 26 April 2017 before the RW-DT show ordinary boreal spring conditions, stratospheric $O_3$ at 16 km is confined north of ~40°N poleward of the subtropical jet as indicated by strong zonal winds. On the morning of 29 April 2017 the RW-DT has streamed mid-latitude stratospheric air (as indicated by $O_3$) toward the equator at 12 – 17 km which is in the process

of being cut off. Quantification and location of the BrO increase in this 12 – 17 km range will help characterize the capability of the integrated retrieval.

A further basis of assessment for the retrieval is the observation of another RW-DT in the vicinity of MLO during the CONTRAST RF01 by the AMAX-DOAS on board the NSF/NCAR GV aircraft. The lower panels of Fig. S2 show the broad
similarity of the RW-DT observed on the afternoon of 11 January 2014 to that on 29 April 2017, with the former being smaller and more completely cut off when observed. The aircraft successfully profiled into the RW-DT allowing for greatly enhanced BrO information content and vertical resolution relative to the MT-DOAS retrievals. Unlike the 29 April 2017 RW-DT, the one observed during CONTRAST was impacted by clouds and required leveraging three separate profiles (see Sect. 3.4 for details).

**3.2 Advances in Spectroscopy and DOAS fit settings**

Fitting of BrO for the integrated tropospheric-stratospheric retrieval required leveraging recent advances in molecular spectroscopy – particularly of $O_3$ and $O_4$ – and managing the spectral cross-correlation of BrO and HCHO. For the levels of BrO we aim to retrieve the optical density of BrO is typically an order of magnitude less than $O_3$ or $O_4$ across different observation geometries making even small changes to the cross-sections important for fitting BrO (Fig. S2). Early fitting
windows for DOAS retrievals of BrO utilized narrower windows covering two or three BrO absorption bands at longer wavelengths (Aliwell et al., 2002) which minimizes spectral interferences by avoiding stronger absorptions by $O_3$ and HCHO. The development of methods accounting for non-linear terms in absorption fitting (Pukite and Wagner, 2016) has helped facilitate the use of wider windows with four bands (Coburn et al., 2016; Koenig et al., 2017; Seo et al., 2019) which are similar to HCHO retrieval windows (Pinardi et al., 2013). The use of wider windows with five BrO bands has been
previously examined for BrO and HCHO (Pukite and Wagner, 2016; Pinardi et al., 2013) and even wider windows have been contemplated (Seo et al., 2019) but ultimately not adopted due to the strong impacts from $O_3$, HCHO, and $SO_2$. We have found that for the low optical densities of BrO (and HCHO) we encounter, rather than avoid these interferences we must characterize them accurately.

The choice of $O_3$ cross-section is critical as the difference between cross-sections multiplied by ZS dSCDs has an optical
density over an order of magnitude greater than that of BrO at twilight (Fig. S3). We employed the same treatment for temperature dependence and orthogonalization (Table S1) for all cross-sections to focus on the cross-sections themselves. Comparison of empirical results reveals the benefits of advances in $O_3$ spectroscopy (Fig. 2). The narrow fit windows used by Aliwell et al., (2002) which specifically avoid strong spectral effects from $O_3$ are found to be generally insensitive to the choice of $O_3$ cross-section as expected. For wider windows, however, significantly smaller BrO columns are measured on
average, with large differences when using different $O_3$ cross-sections. Some variability in retrieved BrO as a function of fitting window is expected due to the different light paths sampled as a function of wavelength (Pukite and Wagner, 2016). Simulating this effect on different BrO bands for ZS observations at twilight finds that the effect is minor for SZA < 90° when accounting for photochemical changes except for the BrO band at ~325 nm (Fig. S4A). The cross-section reported by

Serdyuchenko et al., (2014) yields the most consistent BrO across different windows with deviations from the modeled trend

less than fit uncertainty for most windows apart from the widest. This cross-section is shifted by 3 picometer per the latest recommendations (Gorshelev et al., 2014; Siddans, 2023). Stability for even wider windows will likely require further measurements of $O_3$ cross-sections at the relevant wavelengths, or the application of higher-order corrections for non-linearity. Ultimately, the window starting at 328.5 nm was chosen (Table S1) despite deviating slightly from consistent $O_3$ for ZS data, this is on the basis of constraining $O_4$ and HCHO accurately.

For low elevation angles the choice of cross-section for $O_2$-$O_2$ collision-induced absorption ($O_4$) also changes optical density by more than ten times the total optical density of BrO. We utilize the recently published cross-section by (Finkenzeller and Volkamer, 2022) which significantly changes the $O_4$ band at 344 nm and adds the absorption band at 328 nm. During sensitivity tests for fitting it was found that $O_4$ plays a role in mediating the spectral cross-correlation between BrO and HCHO which we examine further below. This is consistent with previous findings (Pinardi et al., 2013). In order to capture

$O_4$ accurately we first fit $O_4$ in an optimized fitting window (350 – 388 nm) then constrain the $O_4$ fit in the BrO fitting window to a scaled dSCD value. This method has been employed previously assuming a Rayleigh atmosphere for scaling (Koenig et al., 2017). For this work the scaling factor was determined empirically to be 0.80±0.01 by least-orthogonal-distance fitting of the correlation of the unconstrained fits (Fig. S4B). This value is similar to what one might expect for a Rayleigh comparison of optical depth and pathlength of the 360 nm and 344 nm bands $(344/360)^4 = 0.83$ (Wagner et al.,

2004). To better represent the wavelength dependence of the scaling factor, the absorption bands at 344 nm and 328 nm were scaled by the empirical factor, but not that at 360 nm (to account for the wavelength dependent pathlength). Given that the effect is small and the constraint employed emerges empirically from unconstrained fits, one might question the need for it; the key impact of the $O_4$ constraint is to give more consistent fits, which are critical to further constraints discussed below. Further details on the $O_4$ constraint and other constraints are provided in supplement section D.

The $NO_2$ fits retrieved from the BrO fitting window were not consistent with those found in the visible, and indeed seemed non-physical. We suspect that this arises from intensity effects from the changes in illumination at different elevation angles as the fitted intensity offset and Ring cross-sections (which often compensate for such effects in DOAS) show similar patterns. Those effects could be accurately fitted as offsets from changing straylight; notably, the exposure of spectra measured at different EA is dynamically adjusted for consistent saturation in our setup (Coburn et al., 2011). The effect is

too small to be relevant to stratospheric $NO_2$ retrievals, but is relevant to the small $NO_2$ signals measured in the free troposphere for which OA $NO_2$ dSCDs differ from nearby zeniths by less than three times the fit uncertainty. Unlike $O_4$ which is sensitive to only light-path changes, $NO_2$ columns are also impacted by concentration changes which are more difficult to constrain. We first employed $O_4$ and $NO_2$ fits from a more sensitive window in the visible (411 – 490 nm) to invert tropospheric $NO_2$ profiles in the vicinity of instrument altitude. The results from $NO_2$ fitting and resultant profiles will

be described in more detail elsewhere. We then used aerosol optical depth retrieved using the constrained $O_4$ fits described above to model $NO_2$ dSCDs in the BrO fitting window under the assumption of horizontal homogeneity. These $NO_2$ dSCDs were then specified and constrained for further fits of HCHO and BrO.

A major challenging factor in the retrieval of BrO in the troposphere is a remarkable (but coincidental) similarity of the absorption cross-sections of BrO and HCHO when observed at the optical resolutions typical of UV-Vis spectrometers. This similarity in measurement leads to an empirical anticorrelation, the characterization of which is further confounded by chemical coupling of BrO and HCHO via reactions of Br atom with HCHO and other aldehydes, which often correlate with the latter, suppressing BrO formation and creating a chemical anticorrelation. After optimization of the fitting window we believe that this spectral cross-talk is handled by the DOAS fit with the exception of a fast-changing anticorrelation identifiable as opposing changes in BrO and HCHO for sequential measurements of the same viewing geometry, and slow-changing opposing "drifts" in both HCHO and BrO dSCDs. For periods on the order of one hour this "drift" in BrO and HCHO appears to correlate with the overall measured spectral fluxes and/or with small temperature variations of the spectrometer (<0.01 K), however we could not find a consistent correlating parameter nor a definite causal mechanism. As the SZA-dependent variation in HCHO dSCDs is relatively small and there is little prospect without more stable data of retrieving high-altitude HCHO columns, a moving reference analysis was employed for HCHO. The HCHO from the moving reference analysis was used as input to optimal estimation retrievals of HCHO profiles using differential AMFs (dAMFs) as described in Sect. 3.3.2. A posteriori HCHO profiles were then used to compute SCDs using AMFs (not dAMFs). Because the information content in the HCHO retrievals is minimal above 7.4 km, HCHO profiles from CAM-Chem were substituted for these altitudes prior to computing SCDs. The SCD of the fixed reference was then subtracted to yield HCHO dSCDs which were used to constrain the BrO fit. The impact of the HCHO constraint on fits is further detailed in Sect. 3.3.2 as well.

Examining Fig. 2A one might question why the 328.5 – 359 nm window was chosen given that even with the latest cross-sections it deviates from the expected consistency at twilight by more than measurement uncertainty. The benefit of the wider fitting window is to gain access to an additional HCHO absorption feature (peaking between 329 nm and 330 nm; Fig. 2B). The wider window necessitates an order 7 polynomial due to its width and the Huggin's band absorption gradient but at ~45 times the FWHM of the slit function has more than sufficient information for fitting the 17 linear absorption parameters. The combined effect of the constraints on $O_4$, $NO_2$, and HCHO (resulting primarily from HCHO) is to slightly increase BrO dSCDs in general. Without the constraints BrO dSCDs are frequently fit negative for $12° \leq EA \leq 45°$ relative to a zenith reference near noon. For smaller negative values this would indicate surprising profile shapes difficult to reconcile with ZS data; while larger negative values cannot be physically explained. For the 328.5 – 359 nm window application of the fitting constraints for $O_4$, $NO_2$, and HCHO yields positive or (within fit error) zero BrO dSCDs for almost all data, with marked benefits for robust physical dSCDs at the elevated EA off-axis angles (Fig. S4). Leveraging the latest spectroscopy for $O_3$ and $O_4$ and applying constraints on $O_4$, $NO_2$, and HCHO determined from the same MT-DOAS yields robust BrO dSCDs as input to the further retrieval.

### 3.3 Mountaintop Profile Retrievals

The mountaintop BrO profile retrieval, including the DOAS fitting, is summarized in a flowchart in Fig. 3. Broadly, the retrieval proceeds from DOAS fitting to a time-independent profile retrieval, and finally to a time-dependent BrO retrieval. As Fig. 3 makes clear, and as already outlined in Sect. 3.2, intermediate steps are taken. We first examine the aerosol and HCHO profiles retrieved for constraining the BrO DOAS fit on the mornings of 26 and 29 April 2017, then the determination of BrO $SCD_{Ref}$ for the reference spectrum (on April 26, used on both days; see Sect. 2.2); discuss the time-

average retrievals for both days and assess their self-consistency, and finally describe the time dependent profile retrievals for both mornings.

### 3.3.1 Aerosol Profile Retrieval

Aerosol extinction profiles were retrieved for each EA scan to reproduce the scaled $O_4$ dSCDs as described in Sect. 2.3.2. The retrieved aerosol profiles were then used for the HCHO inversion (Sect. 3.3.2) and BrO inversion (Sect. 3.3.3 – 3.3.5).

Prior to the aerosol retrieval an approximate $O_4$ $SCD_{Ref}$ was determined to match the Rayleigh-modeled $O_4$ SCDs for ZS observations. For the base case of Apr. 26 the agreement with the Rayleigh assumption was within $2\times10^{42}$ molec.$^2$ cm$^{-5}$ while for the RW-DT case of Apr. 29 ZS SCDs vary up to $4\times10^{42}$ molec.$^2$ cm$^{-5}$ from this assumption. The EA = 0° $O_4$ SCDs agree with the assumption of a Rayleigh atmosphere to better than 3% in all instances and better than 1% more typically. This level of agreement for EA = 0° is better than fit uncertainty, and highlights a major advantage of MT-DOAS locating

the instrument in an environment where aerosol extinction is sub-Rayleigh extinction. Such low aerosol extinction is detectable to DOAS only at 477 nm, where fit errors are lower than at 360 nm and aerosol contrast is enhanced by reduced Rayleigh scattering (Volkamer et al., 2015). The aerosol optical depth (AOD) above the instrument retrieved by MAX-DOAS at 360 nm ranges from 0.000 – 0.015 for both days which is consistently lower than that found by AERONET direct sun measurements at 340 nm (0.012 – 0.020 on Apr 26; 0.021 – 0.032 on Apr 29) in general agreement given that such low

AOD is close to the limit of detection for AERONET and MAX-DOAS aerosol retrievals. AERONET observations further corroborate an Ångström exponent of ~ 0 between 477 nm and 360 nm. More impactful – but still minor – are the effects from aerosol below the instrument which are observed as a small increase in observed $O_4$ dSCDs for 0° < EA < 90°. The presence of aerosol and/or clouds below the instrument increases multiple scattering at lower altitudes which can add $O_4$ signal given that most of the $O_4$ profile resides below instrument altitude ($O_4$ scales with the square of atmospheric density).

The observed magnitude of this effect at 360 nm is $5.5\times10^{42}$ molec.$^2$ cm$^{-5}$ or 9% in $O_4$ dSCDs at worst and on average less half of that, around the limit of detection. Longer path-lengths and better signal to noise at 477 nm better highlight the effect. We examine this and other factors such as inaccurate EA pointing, and alternative representations such as surface albedo at 477 nm in the supplement (Fig. S5). In brief: inaccurate pointing could explain increases in $O_4$ dSCDs for positive EA but increases discrepancies for negative EA and is not consistent with pointing tests; an elevated increased albedo surface can

also improve agreement for upward looking angles but is at odds with observations in downward and forward geometry; and

finally an increased albedo for a surface at lower altitude can be equivalent in terms of matching $O_4$ dSCDs to the effect modelled by aerosol at low altitude. We chose to use aerosol to represent the aerosol and cloud effects rather than surface albedo, because the latter often necessitates adding similar or more aerosol in any case and we prefer to represent surface albedo closer to its accurate value if possible. The retrieved combined aerosol and cloud optical depth below the site ranges from 0.310 to 0.662 is likely greater than in reality especially as much of it is located below 2 km altitude for which there is minimal sensitivity. Nonetheless, based on the excellent agreement of $O_4$ SCDs to better than fit uncertainty for all EA the (small) aerosol effects are radiatively correct. Retrieval of aerosol profiles especially when so close to Rayleigh conditions and including these secondary effects from multiple scattering below instrument altitude presently exceeds the data collection times by over an order of magnitude, and is the greatest limitation in further application of this analysis.

### 3.3.2 HCHO profile inversions

As described above it was found to be necessary to constrain HCHO for the BrO DOAS fitting; HCHO profiles were therefore retrieved for each EA scan. Since a moving reference analysis is employed for HCHO, corresponding differential weighting functions (dWFs) were used for optimal estimation. The HCHO a priori profile is 600 pptv at the surface with mixing ratio decreasing exponentially with a folding height of 2 km, but not less than 50 pptv. This profile was chosen based on AMAX-DOAS and CAM-Chem (see Sect. 3.4 for details). The optimal estimation uses a 100% a priori covariance for diagonal terms and 0.5 km correlation height for off-diagonal terms below 17.4 km, and 50% covariance and 4 km correlation height above 17.4 km. The retrieved profiles are then combined with WF (non-differential) to obtain HCHO dSCDs as BrO fit constraint.

The retrieved HCHO profiles are summarized in Fig. S6 along with the comparison of measured and a posteriori HCHO dSCDs and SCDs. The effect of constraining HCHO is much greater than those of $NO_2$ and $O_4$ which roughly counteract each other (on average) in their effect on BrO dSCDs; HCHO sensitivities dominate the pattern observed in Fig. S7. The overall effect of constraining HCHO is almost always to increase BrO dSCDs (Fig. S7 data are above the 1:1 line) which given the low SZA zenith reference spectrum is easier to understand physically. The effect of the HCHO constraint – including fast and "drift" components – is $1.5\pm0.9\times10^{13}$ molec. cm$^{-2}$, dominating the overall effect of the constraints which have the same mean and standard deviation. Unlike the other constraints, this is on average almost 2.5 times the fit uncertainty of BrO and dominated by the "drift". Secondary effects are most apparent for larger EA ($\geq12°$) for which dSCDs are "pulled" as a result of the constraints to non-negative, physically meaningful, values relative to the BrO dSCDs of the nearest zenith spectrum. In all instances, these smaller changes are well within the range of uncertainty.They are, however, important to the BrO profile retrieval as they act on relatively small separations between these EA.

Reproducing the dSCDs retrieves independent HCHO profiles below ~6 km, but largely converging to the a priori above this altitude. The DoF for the individual HCHO retrievals ranges from 2.24 to 2.61. This is roughly distributed as one DoF located between 1.9 km and 3.4 km, below the instrument; one DoF located between 3.4 km and 4.4 km near instrument altitude, and the remainder above instrument, with ~70% of the remaining information content below 6 km. Apart from scan

1 on Apr. 26, which has more HCHO, and the latter three scans on Apr. 29, which have decreasing HCHO concentrations, the retrieved HCHO concentration at instrument altitude varies by less than 10% from $2.73\times10^9$ molec. cm$^{-3}$ (161 ppt). In contrast to this relative stability at instrument altitude, HCHO varies more below the instrument, without generally clear trends. Given there is a single DoF below the instrument the profile shape below the instrument should not be interpreted as significant. But the changes in partial columns remain significant. These perhaps reflect dynamic evolution of HCHO in the boundary layer below MLO from biogenic and/or anthropogenic primary and secondary sources. The retrieved HCHO concentrations above the instrument altitude tend to be greater than the 50 pptv a priori. Retrieved HCHO profiles and columns are further compared with aircraft data and other measurements and models in Sect. 3.5 and 3.6.

### 3.3.3 Determination of BrO SCD$_{Ref}$

To obtain SCDs for optimal estimation requires determination of SCD$_{Ref}$. Previous studies have highlighted the impact that SCD$_{Ref}$ can have on BrO retrievals, particularly in regions of low sensitivity. These prior studies had found a need to optimize the distribution of BrO based on a modified Langley plot (Hendrick et al., 2007; Theys et al., 2007) or by iteration until a self-consistent result is obtained (Coburn et al., 2016). A consistent feature of these previous studies has been the connection between upper tropospheric BrO and the BrO SCD$_{Ref}$. In Hendrick et al., (2007) inclusion of the tropospheric BrO column is necessary to retrieve consistent BrO VCDs and thereby a photochemical Langley plot. In Coburn et al., (2016) optimal estimation retrievals with and without SCD$_{Ref}$ were both assessed, with SCD$_{Ref}$ providing additional information content in a broad peak in the upper troposphere. While Theys et al. (2007) examined the connection between SCD$_{Ref}$ and BrO in the upper troposphere less directly, tropospheric AMFs are folded into the parameterization used to derive SCD$_{Ref}$; furthermore, the authors note that the lack of BrO signals near instrument altitude and differences between their retrieved tropospheric BrO VCDs and unpublished results over Nairobi both indicate the tropospheric BrO is likely located at higher altitudes in the troposphere. We adopt a similar approach, assessing different values of SCD$_{Ref}$.

To assess values of SCD$_{Ref}$ we employ photochemical Langley plots as described in Sect. 2.5.1; results from this are summarized in Fig. S8. Because the retrieved profile itself impacts the photochemical AMFs we conduct the time-independent retrieval for various values of SCD$_{Ref}$. This is a fast process because as a trace absorber BrO does not change the WF used for its own retrieval. It is found that SCD$_{Ref}$ = $2.00\times10^{13}$ molec. cm$^{-2}$ recovers itself within uncertainty ($2.01\pm0.09\times10^{13}$ molec. cm$^{-2}$) and has a high $R^2$ value (0.984). Substituting the final retrieved profile shape for the climatological BrO profile retrieves identical results to within the precision of the significant figures. While the $R^2$ value (0.982) and apparent linearity for SCD$_{Ref}$ = $1.50\times10^{13}$ molec. cm$^{-2}$ are similar, it is not self-consistent in that it retrieves a value for SCD$_{Ref}$ which is >3.5σ different ($1.85\pm0.09\times10^{13}$ molec. cm$^{-2}$). Using SCD$_{Ref}$ = $2.50\times10^{13}$ molec. cm$^{-2}$, the photochemical Langley plot is visibly non-linear which is also reflected in a lower $R^2$ value (0.901), and can be more readily rejected on this basis. However, the retrieved value ($2.58\pm0.09\times10^{13}$ molec. cm$^{-2}$) is not significantly different from the previous value, and it might be considered self-consistent. Furthermore, Coburn et al., (2016) employed an iterative approach for SCD$_{Ref}$ using an a posteriori SCD$_{Ref}$ as the a priori which assumes that SCD$_{Ref}$ will tend to converge, however,

the retrieved $SCD_{Ref}$ in this instance diverges further from the optimized value of $2.00\times10^{13}$ molec. cm$^{-2}$ which is used. We find that iterative approaches and goodness-of-fit metrics are not sufficient alone to identify accurate values of $SCD_{Ref}$ rather different values should be surveyed and assessed based on self-consistency and apparent linearity in tandem.

### 3.3.4 Integrated Time-Independent BrO Retrieval

A difference between the twilight BrO SCDs between April 26 and April 29 is apparent by simple examination (Fig. 4). On April 26 BrO SCDs are observed to decrease monotonically as the sun rises from values starting from almost $2.5\times10^{14}$ molec. cm$^{-2}$, whereas on April 29 they at first increase from lower values of ~$1.7\times10^{14}$ molec. cm$^{-2}$ before an umkehr around SZA = 90° then decreasing by less to a relatively high BrO SCD at SZA = 75° (Fig. 4). These differences together indicate additional BrO above the site on April 29 relative to April 26 located below the mean scattering altitude for SZA ≥ 90° (~30 km). Zenith SCDs at lower SZA, however, are similar on both days necessitating additional differences; these are ultimately determined to be dynamics-driven profile changes over the course of the measurement period on April 29. On both days intermediate angles (3 – 30° EA) are often elevated relative to lower elevation angles, suggesting BrO is not enhanced at instrument altitude, but aloft in the free troposphere. The residuals of the time-independent retrieval on April 26 are within 2% of the mean slant column fit uncertainties and typically are smaller Comparing the residuals (Fig. 4) to these mean uncertainties for April 26: the overall difference is very slightly larger for all observations 6.2 vs $6.1\times10^{12}$ molec. cm$^{-2}$, but smaller (8.1 vs $8.3\times10^{12}$ molec. cm$^{-2}$) for ZS data, and (3.6 vs $4.1\times10^{12}$ molec. cm$^{-2}$) for OA data. For April 29, by contrast, the comparison for ZS data is even better, 7.9 vs $9.1\times10^{12}$ molec. cm$^{-2}$; while OA data compare less favorably, but still within 60% (7.1 vs $4.4\times10^{12}$ molec. cm$^{-2}$) in turn driving poorer performance overall (7.1 vs $6.6\times10^{12}$ molec. cm$^{-2}$). Examination of the time series of the retrieval residuals from the time-independent retrieval (black symbols Fig. 4d background) reveals a clear pattern: for ZS data residuals scatter around zero, increasingly converging as more photons improve signal to noise, rather suddenly at SZA=75° residuals are systematically positive across EA, then decrease approximately linearly to be systematically negative at SZA=40°. This is indicative of changes in BrO above those OA measurements are most sensitive to but below those ZS measurements are most sensitive; potentially consistent with an evolving RW-DT event. While it is clear that the retrieval, especially on April 29, can be further improved (as is done below in Sect. 3.3.5) the magnitude of residuals indicates the results of the time-independent retrieval are sufficiently accurate to examine.

Examination of the retrieved time-independent profiles and the associated AVKs (Fig. 5) and particularly DoF (Table 1) can better quantify the major differences in the BrO profile brought about by RW-DT. Examination of the AVKs and column sensitivities reveals the two regions of high information content at the bottom and top of the profile. Below ~7.4 km, the retrieval has three DoF with reasonable precision of altitude: 1.9 – 3.4 km (1.1 DoF), 3.4 – 4.4 km (1.1 DoF), and 4.4 – 7.4 km (1.0 DoF). A local minimum in BrO is observed near instrument altitude on both days, however, the altitude regions each possess a single DoF in aggregate; considering this, the BrO at instrument altitude is not significantly less than below the instrument on April 29. The more striking difference between the two days is above the instrument, in the 4.4 – 7.4 km

range, which on April 26 peaks at over twice the concentration at instrument altitude. While on April 29 the BrO concentration decreases on average broadly in line with air density (see also Fig. 5). There is no obvious connection between a RW-DT event and this altitude range, suggesting the observed differences are coincidental, or that something more subtle in the atmospheric dynamics or retrieval is responsible. The column sensitivity (Fig. 4) reveals the other region of significant information content (1.5 DoF) from ZS data concentration in the stratosphere above 17.4 km. The retrieved stratospheric

BrO profile on April 26 is broadly consistent with those previously measured by balloons in the topics (Pundt et al., 2002; Dorf et al., 2008), peaking between 20 and 25 km at $1.2\pm0.1$ molec. $cm^{-3}$. The effect of the RW-DT on the 29[th] is immediately apparent as the stratospheric BrO profile is shifted to lower altitude, and with a larger total column, both better resembling midlatitudes. It is notable, that the decrease in BrO concentration is greater than the decrease of air density above 25 km, which would indicate a BrO concentration decrease which we do not believe has been previously observed.

However, as the AVK reveal, information content in the stratosphere is imprecise for the attribution of altitude to BrO signal and information content is especially limited at these highest altitudes; therefore we suspect that this decrease in mixing ratio is likely false, and the BrO peak at lower altitude is slightly overestimated in a manner consistent with the AVK and that sensitivity above ~30 km is limited though better signal to noise for SZA > 90º could address this.

Accounting for additional sources of uncertainty in the measurement covariance ($S_\varepsilon$) highlights the robustness of the

620 retrieval. Accounting for the uncertainty in the BrO absorption cross-section (~10.5%; Fleischmann et al., 2004) as well as the change in BrO dSCD from the $O_4$ and $NO_2$ constraints and the non-drift component of the HCHO fit constraint (order of $10^{12}$ molec. $cm^{-2}$) fundamentally lowers the signal to noise, leading the retrieval to generally trend slightly toward the a priori and be smoothed over (Fig. S9). The "drift" observed in the HCHO-BrO cross-talk is clearly an instrumental effect, however, we account for it in a further sensitivity study to further probe the most robustness of the retrieval. The effect on

Apr 26 is, as expected, minimal and while more than one DoF is lost on Apr 29, the change in VCD is small and the increased BrO in the upper free troposphere is still retrieved (more detail in supplement section D). Even accounting extremely conservatively for uncertainty, the retrieval already improves on previous BrO retrievals.

Below ~1.9 km the sensitivity of the MT-DOAS is not only limited but negligible (<0.1 DoF) and constitutes a null space for retrieval. This leaves one region of the atmosphere between the lower troposphere (1.9 – 7.4 km; 3.2 DoF) and the

630 stratosphere (>17.4 km; 1.5 DoF): the upper troposphere (7.4 – 17.4 km) is not readily retrieved using OA or ZS data alone, a local but nonzero minimum in column sensitivity in the upper troposphere. The joint leverage of ZS and OA data allows 0.96 – 0.97 DoF to be retrieved between 7.4 – 10.4 km, clearly revealing the impact of the RW-DT even on the BrO concentration in the upper troposphere. On April 26, the BrO profile decreases with altitude to a minimum, not significantly different from zero, at 11.4 – 13.4 km. The increase above this altitude while still in the troposphere is consistent with

635 previous profiles on mobile platforms (Pundt et al., 2002; Dorf et al., 2008; Koenig et al., 2017) During the RW-DT event on April 29, however, there is instead a BrO maximum in the upper troposphere presumably resulting from the injection of air from the stratosphere with a much higher BrO mixing ratio. This stratospheric intrusion on April 29 is captured with an

independent DoF from the synergy of ZS and OA observation indicating it is independently significant and not the result of other differences in the profile.

### 3.3.5 Time-Dependent BrO Retrieval

The SCD retrieval residuals on April 29 show a clear temporal pattern, particularly for OA data (Fig. 4); as noted in Sect. 3.3.4 these are the only subset of data for which the retrieved residuals exceed the average fitting uncertainties. Given that the photochemical variation in $Br_y$ partitioning is already reflected in the air mass factor calculations, we attribute this to additional time dependent changes in BrO (and possibly $Br_y$). The set up of the time-dependent retrieval is given in Sect. 2.5.3. Sensitivity studies used to determine the values ultimately used are described in the supplement section C. In brief, in Eq. 3, $f$(t) was set to be a ramp function with value zero for SZA < 70° increasing linearly with time to a value of one for remaining data based on time dependent $O_3$ VCD retrieval for ZS data (Fig. S10). Four atmospheric layers ($\mathbf{W^L}$) for the lower, middle, and upper troposphere, and for the stratosphere based on CAM-Chem modelling for Apr. 29 (Fig. S11). The retrieved profile is sensitive to the choice of a priori profile ($\mathbf{x_0}$; Fig. S11) with a western Pacific profile even having slightly smaller average retrieval residuals but extreme gradients in BrO at low altitude which were deemed likely unphysical. The retrieval was found to be highly sensitive in an unpredictable manner to the choice of a priori values for $\mathbf{C^L}$ which were chosen based on selecting consistent results when varying the BrO a priori profile ($\mathbf{x_0}$ a priori) and for small changes in the $\mathbf{C^L}$ a priori (Fig. S13) from a systematic survey of choices. Settings optimized for April 29 were also applied to April 26, which was found to be less sensitive to the choice of a priori. 1.7 and 1.8 time-dependent DoF were retrieved for April 26 and April 29 respectively.

RW-DT events involve the movements of mid-latitude air toward the tropics which might be expected to increase the BrO VCD, however, the difference in BrO VCD is not significant whether examined at SZA = 70° or at the minimum SZA: base case (SZA = 70° to SZA = min.;2.2 to 2.3 ± 0.2)×$10^{13}$ molec. cm$^{-2}$ vs RW-DT (2.6 to 2.4 ± 0.3)×$10^{13}$ molec. cm$^{-2}$. The stratospheric BrO VCD differs by 5 – 10% between the two days, (1.46 to 1.47 ± 0.08)×$10^{13}$ molec. cm$^{-2}$ in the base case, and (1.61 to 1.55 ± 0.08)×$10^{13}$ molec. cm$^{-2}$ for the RW-DT (Fig. 6), broadly consistent with the 7% difference in the $O_3$ VCDs for SZA < 70° (Fig. S10). Given the observed increase in $O_3$ VCDs during April 29, it could be expected that the observed decrease in stratospheric BrO VCDs over the morning of April 29 must be compensated by a tropospheric increase resulting from the RW-DT, however BrO SCDs are actually lower than expected by the time-independent retrieval (Fig. 4). Consistent with this the time-dependent tropospheric BrO VCD decreases from (1.01±0.14)×$10^{13}$ molec. cm$^{-2}$ to (0.85±0.17)×$10^{13}$ molec. cm$^{-2}$. Counterintuitively, given the expectation that RW-DT is typically conceived to inject BrO into the troposphere, BrO decreases by (0.15±0.17)×$10^{13}$ molec. cm$^{-2}$ from (1.01±0.14)×$10^{13}$ molec. cm$^{-2}$ in the RW-DT case mostly in the mid to lower-FT (>80% of the change), the altitudes furthest from the stratosphere. That tropospheric BrO increases slightly, (0.13±0.16)×$10^{13}$ molec. cm$^{-2}$ over (0.70±0.14)×$10^{13}$ in the base case demonstrates the capacity of the retrieval to produce such an increase when reflective of the underlying data. The small to negligible change in the stratosphere and upper FT for the RW-DT has 0.75 total DoF. This suggests that while the $O_3$ VCD provides a reasonable

estimate of the change in stratospheric BrO VCDs between the two days, it does not readily predict changes during the RW-DT event. Nonetheless, the time evolution of the $O_3$ VCDs has been found to roughly correspond to changes in BrO, suggesting there is some connection between the changes. On both days the largest changes are observed in the lower atmosphere (< 7.4 km). On April 26 the change is primarily in the 4.4 – 7.4 km range where BrO increases from 0.7±0.1

675  pptv to 0.9±0.1 pptv which is very slightly compensated by minor increases at lower altitudes. Changes on April 29 are roughly the opposite (0.5±0.1 pptv to 0.3±0.1 pptv) and more consistent with the changes at lower altitudes (0.46±0.03 pptv to 0.29±0.04 pptv at 3.4 – 4.4 km; and 0.32±0.06 pptv to 0.20±0.07 pptv at 1.9 – 3.4 km). Even if the observed increase in $O_3$ is in or near the stratosphere, it would be expected to slightly lower $J_{BrO}$ via increased UV absorption increasing the BrO:Br ratio holding other chemical conditions constant. The observed decrease in BrO in the lower FT therefore points to

either a decrease in $Br_y$ or chemical changes lowering the BrO:$Br_y$ ratio, which have some link to the changes in $O_3$ VCDs.

## 3.4 Aircraft Profile Retrievals of HCHO and BrO

Aircraft observations of a second RW-DT event in the vicinity of MLO by the CU-AMAX-DOAS provide an opportunity to assess the retrieval leveraging the greater sensitivity and precision of altitude available aboard a mobile platform. As has been previously noted, RW-DT are often accompanied by widespread and complex cloud fields presenting challenges for

DOAS retrievals due to the radiation fields; this was the case on January 11, 2014, however, two descents and an ascent over Hawaii together provide a near complete profile sampled twice with clear line of sight. The relevant flight path is summarized in Fig. 7. BrO profiles from the CONTRAST campaign have been previously reported in (Koenig et al., 2017). After conducting sensitivity studies we have made some minor adjustments to the BrO fit settings to better match those used for MLO in this work summarized in Table S2, with further details in the supplement.

The retrieved BrO and HCHO profiles are summarized in Fig. 8. In the boundary layer below ~2 km altitude significant heterogeneity is observed. The rapid changes in observed signals around the missed approach at Kona airport (KOA) present challenges for the optimal estimation leading to strikingly different concentrations at neighboring altitudes and between RF01-06 and RF01-07 (Fig. S15 and S16). Comparison with the interpolated BrO profile from CAM-Chem shows that the model underestimates the local concentrations, however, the shading showing the range of concentrations over the Hawaiian

Islands shows that the model does include instances of high BrO concentrations in the MBL, and is only missing the precise location of these in the vicinity of KOA. A number of points retrieved in the HCHO profile for RF01-07 in the boundary layer – but not for RF01-06 – are outside the measured range of in-situ observations by ISAF indicating there might have been very low HCHO air somewhere in the vicinity but not along the flight track, although the large propagated uncertainty in the optimal estimation indicates limitations of the retrieval also likely play a role. Retrieved BrO mixing ratios in the FT

from 2 km to 9.5 km oscillate around ~0.5 ppt with a minor local maximum around 5 – 6 km captured in all profiles (see also Fig. S15). CAM-Chem generally captures the broad picture of BrO in the FT, but appears to misplace the vertical extent of RW-DT stratospheric intrusion down as low as 8 km, which is not observed. Instead, the stratospheric intrusion is observed to start at 10 – 12 km where BrO is observed to increase to over 2 ppt with the intensity again mildly underestimated by the

model. In the lower FT, both the ISAF and AMAX-DOAS find that HCHO mixing ratios are not only higher than predicted
by CAM-Chem, but outside the regional range in the model up to roughly the altitude of MLO. Through the altitude range of
4 – 8 km the measurements remain in general agreement, with the ISAF possessing greater precision, the model is within
uncertainty over this range but still systematically slightly low. The measurements again depart from CAM-Chem
predictions in the stratospheric intrusion where the model predicts <50 ppt HCHO, but both instruments observe ≥100 ppt.
This discrepancy was not generally observed by ISAF during flights over the central Pacific but is more typical of the
western Pacific (Anderson et al., 2017).

## 3.5 Retrieved Profiles in Context

The CONTRAST RF01 profiles offer only a limited snapshot over the central Pacific, but nonetheless offer an opportunity to
compare to prior aircraft measurements over the western and eastern Pacific. Examining the profiles over the altitude range
the MT is sensitive to (Fig. 9), the RF01 retrieved profile is broadly consistent with the eastern Pacific in the lower FT. This
is particularly notable because, as noted above, BrO was observed in the MBL similar to the western Pacific, but this is not
found to propagate up to the lower FT as it does there. This might reflect lower convective intensity relative to the western
Pacific, but might also be a reflection of limited sampling and statistics. Through the range of 4 – 9 km all three regional
profiles are approximately 0.5 ppt BrO, indicating this might reflect a broadly applicable Pacific background with some
uncertainty. Above this altitude, the RF01 profile is again in general agreement with the average profile observed over the
eastern Pacific. Keeping in mind that the observed profile is during a RW-DT, this suggests that such events, and isentropic
transport from the stratosphere more generally which are common in the region (Wernli and Sprenger, 2007; Funatsu and
Waugh, 2008), are likely contributors to the observed increase in BrO in the upper FT observed over the eastern Pacific.
Intriguingly, the decrease in BrO lower in the FT observed at the same time may reflect the effects of dynamics which are
broadly coupled to Rossby wave breaking (Funatsu and Waugh, 2008; de Vries, 2021) and it is possible these dynamics can
also contribute to the relatively low BrO concentrations at lower altitudes observed over the eastern Pacific.
Comparing the MT and aircraft profiles from this work (Fig. 9), BrO is consistently ~ 0.5 ppt, again generally suggesting this
might be a regional average within some bounds of altitude. We apply the AVK from the MT-DOAS retrieval to the
AMAX-DOAS profile to better compare how the two profiles would appear on the same instrument. Interestingly, the
maximum of ~1.0 ppt BrO observed around 6 km altitude on Apr. 26 is broadly similar to the upper range of a local
maximum observed over the western Pacific, suggesting it might arise from similar processes over the western Pacific. The
low BrO observed above 9 km on Apr. 26 is also broadly consistent with the western Pacific although again more marked
than average. In contrast to the RW-DT-impacted profiles, Apr. 26 more closely resembles the western Pacific. This suggests
that variability in BrO profiles, over the central Pacific and perhaps more broadly, might be partly driven by meteorological
conditions which occur and are sampled with different frequency regionally. Intriguingly, the aircraft profile observes a
similar BrO minimum between 6 km and 9 km as on Apr. 29. As previously noted this altitude range does not obviously
connect to the RW-DT events observed. With only two samples it is possible this is merely coincidental, and it should be

noted that applying the AVK for Apr. 29 to the aircraft profile eliminates this feature. However, we posit that local maximum in BrO observed on Apr. 26 and over the western Pacific, and this minimum might have their origins in convective transport (and perhaps the lack thereof) which is known to be systematically perturbed prior to and during RW-DT events. Examining the stratospheric intrusion above ~12 km associated with the RW-DT, the two profiles are remarkably similar when accounting for the AVK. Consistent with the CAM-Chem model prediction that the RW-DT on Apr. 29, 2017 is greater in intensity than on Jan 11, 2014 (Fig. S2) more BrO is observed in the upper FT from the MT when accounting for AVK. This demonstrates that the MT DOAS is sensitive to BrO in the upper FT and can detect the impact of a RW-DT with AVK likely accurately capturing the limitations of the retrieval (Rodgers and Connor, 2003). These limitations as represented by the AVK should be considered when interpreting the data as the comparison with the aircraft profile demonstrates the limited altitude precision of the MT retrieval.  It is likely that BrO enhancements are more limited in vertical extent than captured by MT-DOAS, but correspondingly more intense.

## 4 Conclusions and Outlook

MT-DOAS is well suited for measuring trace gas profiles up to 35 km altitude, able to retrieve more than one degree of freedom in each of below the instrument, near instrument altitude, the free troposphere, and the stratosphere. We further highlight a number of lessons learned and prospects for extension to other work.

### 4.1 Lessons Learned

Development of the retrieval method highlighted a number of features of the MT-DOAS retrieval of BrO:

- The placement of the MT-DOAS above most atmospheric aerosol meant that aerosol at and above instrument altitude were minimal to almost negligible, however, it also revealed the need to represent aerosol and cloud radiative effects below the instrument.

- Recent advances in the molecular spectroscopy of $O_3$, and $O_2$-$O_2$ collision induced absorption (Serdyuchenko et al., 2014; Finkenzeller and Volkamer, 2022) greatly improve consistent spectral fitting of BrO and HCHO dSCDs from solar stray light spectra measured in the ZS and OA geometries .

- Nonetheless, spectral cross-correlation between BrO and HCHO still requires active management and imposing constrains consistent with prior findings (Pinardi et al., 2013; Pukite and Wagner, 2016; Seo et al., 2019).

- Combining the ZS- and OA-based retrievals of stratospheric and tropospheric BrO respectively has a synergistic benefit of gaining roughly one degree of freedom.

- Time-dependent variables were needed to represent changes in $Br_y$ faster than one day. When added to the purely spatial variables typically used for optimal estimation they allowed for a consistent inverse Bayesian treatment of the data, and agreement with observations, however, the sensitivity of the new variables is still incompletely understood.

Additionally, knowledge of atmospheric BrO was extended by the first measurements over the central Pacific by MT-DOAS and AMAX-DOAS.

- The central Pacific BrO profile is generally consistent with the western Pacific in the base case, however, the RW-DT case highlights the likely contribution of such events to elevated BrO at high altitude over the eastern Pacific.
- Unexpectedly, given that RW-DT are typically understood primarily as injections of stratospheric air, short-term changes in BrO (faster than one day) were found to occur mostly lower in the troposphere.

## 4.2 Prospects for Further Development and Application

The methods developed here can be built upon and extended to other remote sensing retrievals. Key needs for further development include:

- Overcoming the key limitation imposed by computational time of the RTM for aerosol retrieval, despite low aerosol conditions. Consolidating the RTM computations within a single model and perhaps using look up tables if necessary could help in this direction.

- The only remaining null space for the retrieval is below 1.9 km. Coordination of MT-DOAS with a MAX-DOAS is one potential long-term solution to this, but there may be others.

The methods developed here can also be extended to different trace gases using MT-DOAS. The synergistic benefit of combining the ZS and OA retrieval is greatest where the stratospheric and tropospheric partial columns are comparable (for BrO less than 1:2 either way); and the benefit of time-dependent extensions to optimal estimation is greatest for species

where there is extensive nonphotochemical heterogeneity. Species to be targeted in an approximate order of suitability are:

- $NO_2$: which in an urban environment has tropospheric and stratospheric partial columns within the same 1:2 ratio and even in the remote atmosphere still frequently has a tropospheric to stratospheric partial column ratio of 1:6 or better. Furthermore, the retrieval of $NO_2$ can be conducted at multiple wavelengths which especially at low AOD can be used for ranging (Ortega et al., 2015; Dimitropoulou et al., 2022); employing this ranging approach in

    conjunction with the time-dependent methods introduced here holds potential to further develop and assess mutli-dimensional optimal estimation approaches.
- HCHO: might be considered as a target because it is expected to be fitted and retrieved in the course of BrO retrievals in any case. In regions of low HCHO, the stratospheric column on the order of $1 \times 10^{14}$ molec. cm$^{-2}$ can be as much as 10% of the total column, however, it is typically less than this. Tropospheric HCHO does, however,

    have complex drivers from both photochemistry and transport which might benefit from a time dependent approach.
- IO and CHOCHO: are expected to have even less favorable conditions in terms of the relative proportion of their stratospheric partial columns, but the comparability of tropospheric and stratospheric columns specifically might not be the key criterion as both have significant columns in the free-troposphere. Both also have significant tropospheric heterogeneity to unravel.

- O$_3$: in contrast to most other species measured by DOAS has its VCD dominated by the stratosphere with the troposphere accounting for a single digit percentage of the total column typically. Nonetheless, a full atmosphere retrieval might be possible for the Chappuis bands which have longer pathlengths and better signal to noise for OA geometries, or perhaps by leveraging both the Chappuis and Harley bands together.

*Code Availability.* A set of functions used for the optimal estimation written for Igor 7 as well as the code used for the final retrieval are archived at https://doi.org/10.5281/zenodo.11570073.

*Data Availability.* MT-DOAS data and associated RTM data products are archived at https://doi.org/10.5281/zenodo.8337857. The AMAX-DOAS BrO data are available from the CONTRAST data archive: http://data.eol.ucar.edu/master_list/ ?project=CONTRAST. The CONTRAST data set is open for use by the public, subject to the data policy: https://www.eol.ucar.edu/ content/contrast-data-policy.

*Author Contributions.* TKK, FH, MVR, and RV conceptualized this work and developed the methodology. TKK, CFL and RV conducted measurements. TKK, FH, CFL, and MVR conducted data analysis. DK provided model data and analysis tools. TKK visualized data with contributions from all authors. TKK and RV wrote the manuscript with contributions from all authors.

*Competing interests.* The authors declare that they have no conflict of interest.

*Acknowledgements.* This work was funded by the National Science Foundation (NSF; AGS-2027252, AGS-1649147, and AGS-1951514). CONTRAST was funded by the NSF (AGS-1261740, and AGS-1620530). Mauna Loa Observatory is a National Oceanic and Atmospheric Administration (NOAA), Earth System Research Laboratory (ESRL) Global Monitoring Laboratory (GML) facility. CAM-Chem is a component of the Community Earth System Model, supported by the National Science Foundation (NSF). We would like to acknowledge high-performance computing support from Cheyenne (doi:10.5065/D6RX99HX) provided by NCAR's Computational and Information Systems Laboratory, sponsored by the NSF. We thank the NOAA staff at Mauna Loa Observatory for assistance with the on-site maintenance and calibrations on the MT-DOAS instrument, especially Paul Fukumura. We further thank Barbara Dix for initial set up of the MT-DOAS at Mauna Loa and preliminary discussions toward this work, as well as the CONTRAST team, including pilots, technicians, forecasters, and scientists on the aircraft and on the ground. TKK and RV thank Thomas Hanisco, Glenn Wolfe, and Daniel C. Anderson for use of the CONTRAST ISAF data. The GV aircraft was operated by the National Center for Atmospheric Research's (NCAR) Earth Observing Laboratory's (EOL) Research Aviation Facility (RAF).

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

**Table 1. Retrieved Degrees of Freedom Summary**

| Time-independent Layers (km) | Apr. 26 DoF | Apr. 29 DoF | Time-dependent Layers (km) | Apr. 26 DoF | Apr. 29 DoF | Total DoF Apr. 26 | Total DoF Apr. 29 |
|---|---|---|---|---|---|---|---|
| 1.9 – 3.4 | 1.12 | 1.13 | 0 – 6 | 0.59 | 0.61 | 1.32 | 1.33 |
| 3.4 – 4.4 | 1.06 | 1.06 | | | | 1.26 | 1.26 |
| 4.4 – 7.4 | 0.98 | 0.97 | | | | 1.17 | 1.17 |
| 7.4 – 17.4 | 0.97 | 0.96 | 6 – 10 | 0.91 | 0.46 | 1.92 | 2.09 |
| | | | 10 – 17.5 | 0.04 | 0.67 | | |
| 17.4 + | 1.47 | 1.44 | 17.5 + | 0.14 | 0.08 | 1.61 | 1.52 |
| Total | 5.60 | 5.57 | Total | 1.68 | 1.81 | 7.28 | 7.38 |

1120

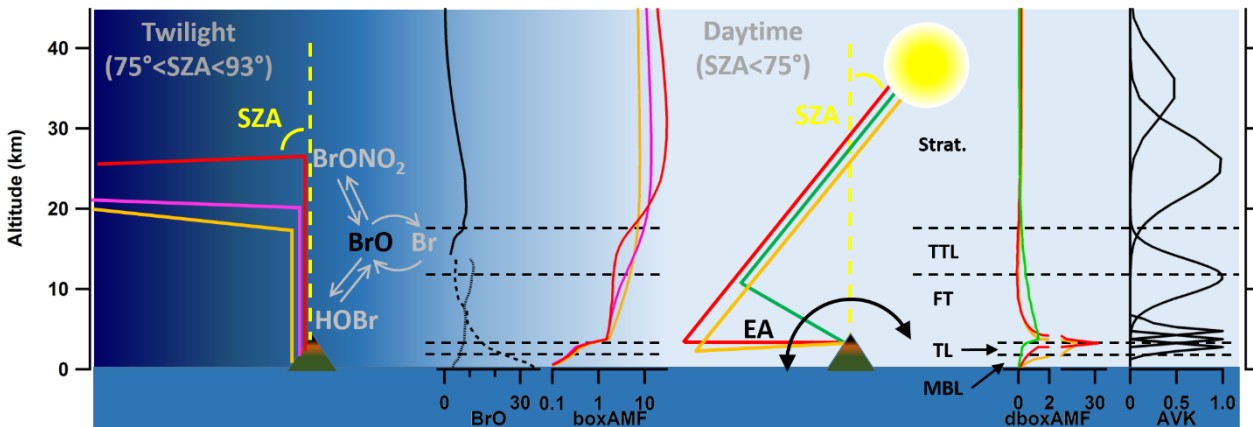

**Figure 1: Principals and components of integrated BrO retrieval.**

From left to right. At twilight, high solar zenith angle (SZA), measurements are taken in a zenith-sky (ZS) geometry and must account for the rapid photochemical changes to bromine in both time and along the light-path. BrO mixing ratios are shown for the stratosphere (solid line), western Pacific (long dashes), and eastern Pacific (fine dashes) next to the box air mass factors (boxAMF) showing which altitudes measurements at different SZA (matching color to left) are sensitive to (red SZA=92°, pink SZA=88°, orange SZA=84°). These peak strongly in the stratosphere and more mildly in the tropical transition layer (TTL) and upper free troposphere (uFT). When SZA<75° the instrument varies the elevation angle (EA) viewing geometries relative to the horizon (OA). The differential boxAMF relative to a ZS measurement at the same times is shown for different EA (orange -3°, red 0°, green 30°). These peak below the instrument in the transition layer (TL), at instrument altitude (lower FT to TL) and lower FT. Combining these sensitivities allows for 5.5-6 independent measurements conceptually shows as averaging kernels (AVK) on the right. Far above the instrument almost three independent partial columns are retrieved, almost two in the stratosphere (primarily from ZS-DOAS data), and one in the upper FT and TTL through ZS-DOAS and OA-DOAS synergy. Near instrument altitude there is relatively fine vertical resolution (from OA-DOAS) and three mixing ratios are retrieved.

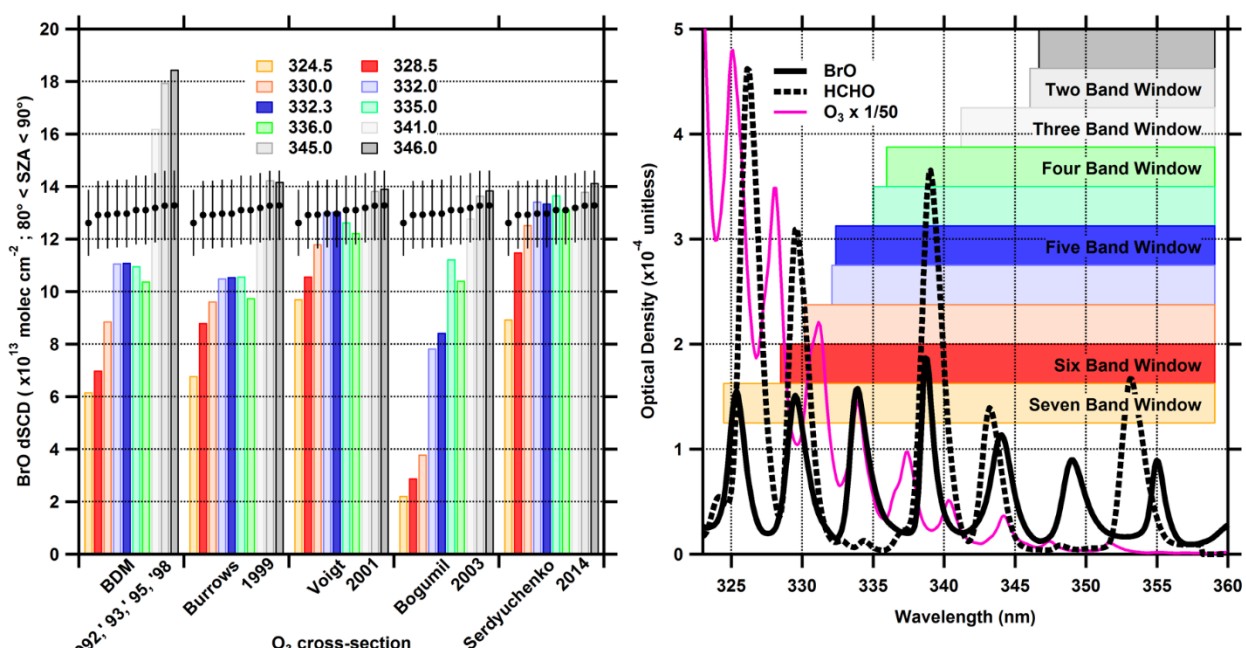

**Figure 2: Advances in O₃ molecular spectroscopy, and the effect on a robust BrO fit.**

Left: average BrO dSCD fits for twilight data in different spectral fitting windows when using different O₃ cross-sections. Right: representative optical densities as observed at MLO for SZA=70° and EA=0°. Colored bars indicate DOAS fitting windows for which the short wavelength edges are given in the legend (the long wavelength edge is always 359 nm) and graphically shown on the right to indicate the number of BrO bands included. Bars on the left compare the average measured BrO dSCD (80°<SZA<90°) with that expected due to changes in light-paths for the different windows (due primarily to O3; black circles are scaled to match the average from windows with five BrO bands or fewer), error bars indicate the average DOAS fit uncertainty in the same SZA range.

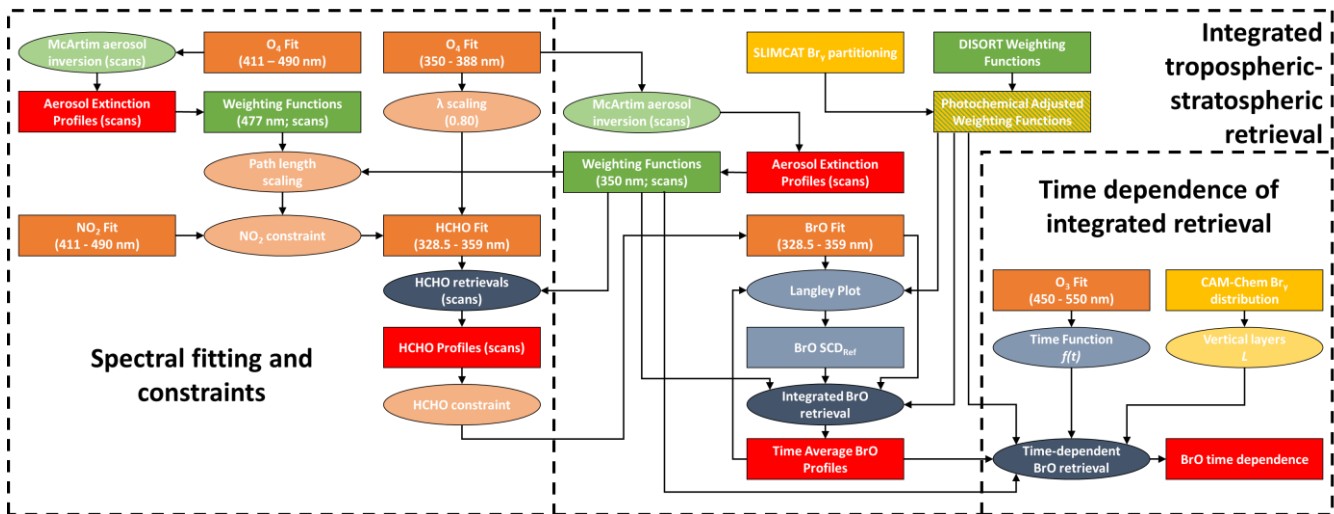

**Figure 3: Summary of MT-DOAS BrO retrieval methods**

Rectangles identify data products while ellipses indicate processing tools. In orange spectral fitting processes and products, in green radiative transfer modelling tools and products, in yellow chemical data products and tools, in blue optimal estimation and other retrieval methods, in data products reported in this work.

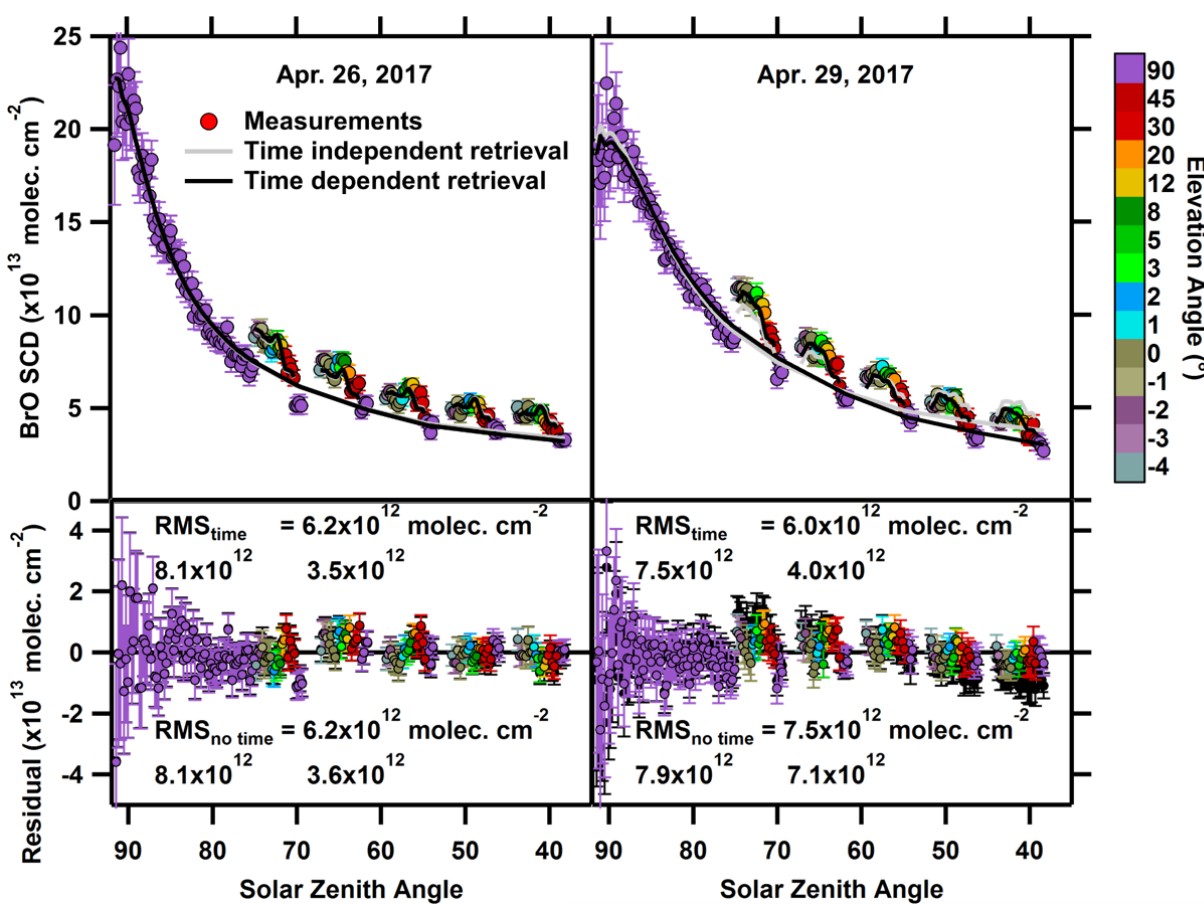

**Figure 4: Comparison of measured and retrieved BrO SCDs.**

Top: Retrieved reproduction of measured BrO SCDs for the mornings of April 26 (left) and 29 (right), 2017. Measurements are color coded by EA with error bars showing the DOAS fitting error (the $SCD_{Ref}$ = 2.0×1013 molec. cm-2 is included) against SZA (sunrise is on the left of each window). The grey lines show the retrieval for the zenith data, and for each EA scan normalized to SZA 70° without time dependence accounted for. The black line shows the retrieval allowing for a time dependence of the BrO profile (the difference is almost imperceptible for April 26). The residuals (= retrieved – measured) are shown on the bottom for the time independent retrieval (solid black) and the time dependent retrieval (color coded). Also listed are the root-mean-square (RMS) of the residuals for each retrieval, as well as for subranges of SZA>75° and SZA<75°. The primary benefit of including the time dependence is to eliminate the slope apparent in the residuals for SZA<75° on April 29.

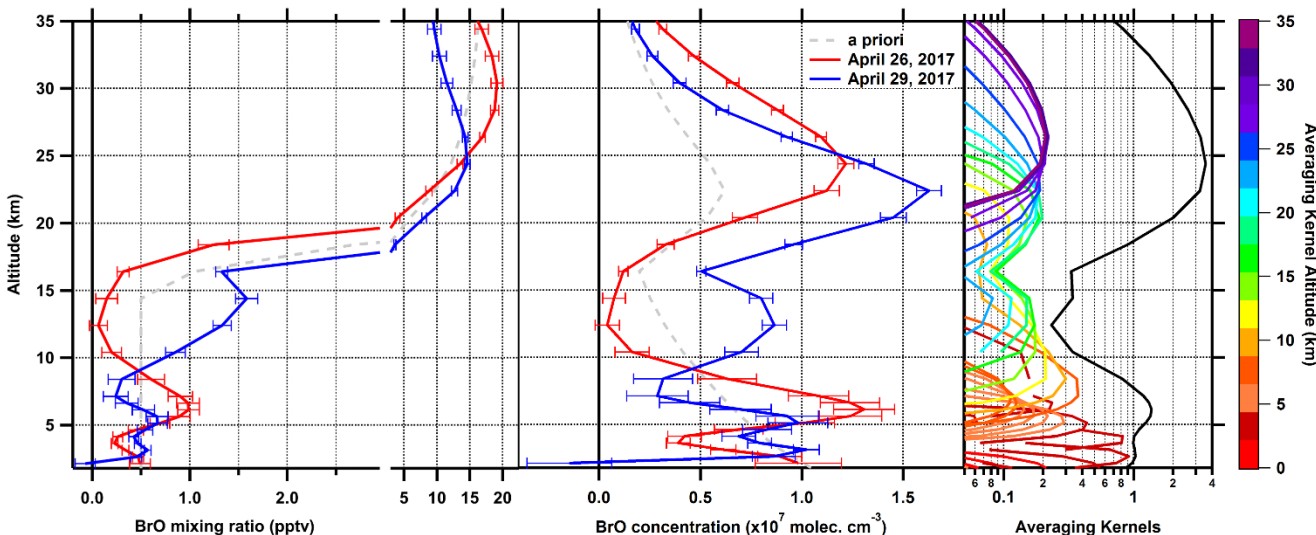

**Figure 5: The integrated BrO profile retrieval.**

Left: Bro mixing ratio profiles at SZA=70° for the mornings of April 26 and April 29, 2017, utilizing the default a priori (grey dashed line). Middle: The corresponding BrO concentration profiles at SZA=70°. Right: Averaging Kernels (for April 26) at the kernel altitude (color coded), and the column sensitivity (black line, the sum of all kernels at a given altitude). The vertical resolution is best near instrument altitude, and decreases in the uFT and stratosphere; the column sensitivity drops as low as ~0.3 in the altitude range of 10-17 km, but does not drop lower.

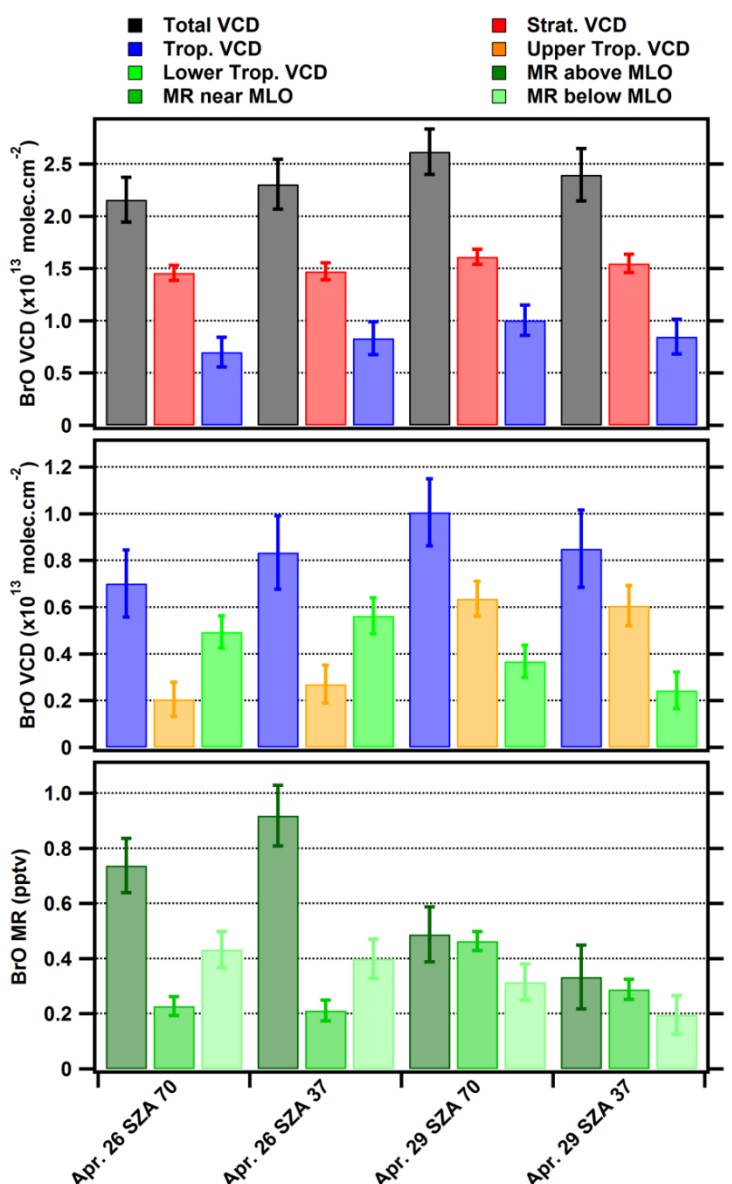

**Figure 6: Retrieved BrO columns and mixing ratios and time evolution.**

Top: the total, stratospheric (17.4 – 53.4 km), and tropospheric (1.9 – 17.4 km) BrO VCD, the error bars indicate the total uncertainty; middle: the tropospheric BrO VCD (repeated) is broken into upper FT (7.4 – 17.4 km) and lower FT (1.9 – 7.4 km). Bottom: three mixing ratios retrieved in the lower FT are averaged above the instrument (3.9 – 7.4 km), near the instrument (3.4 – 3.9 km), and below the instrument (1.9 – 3.4 km).

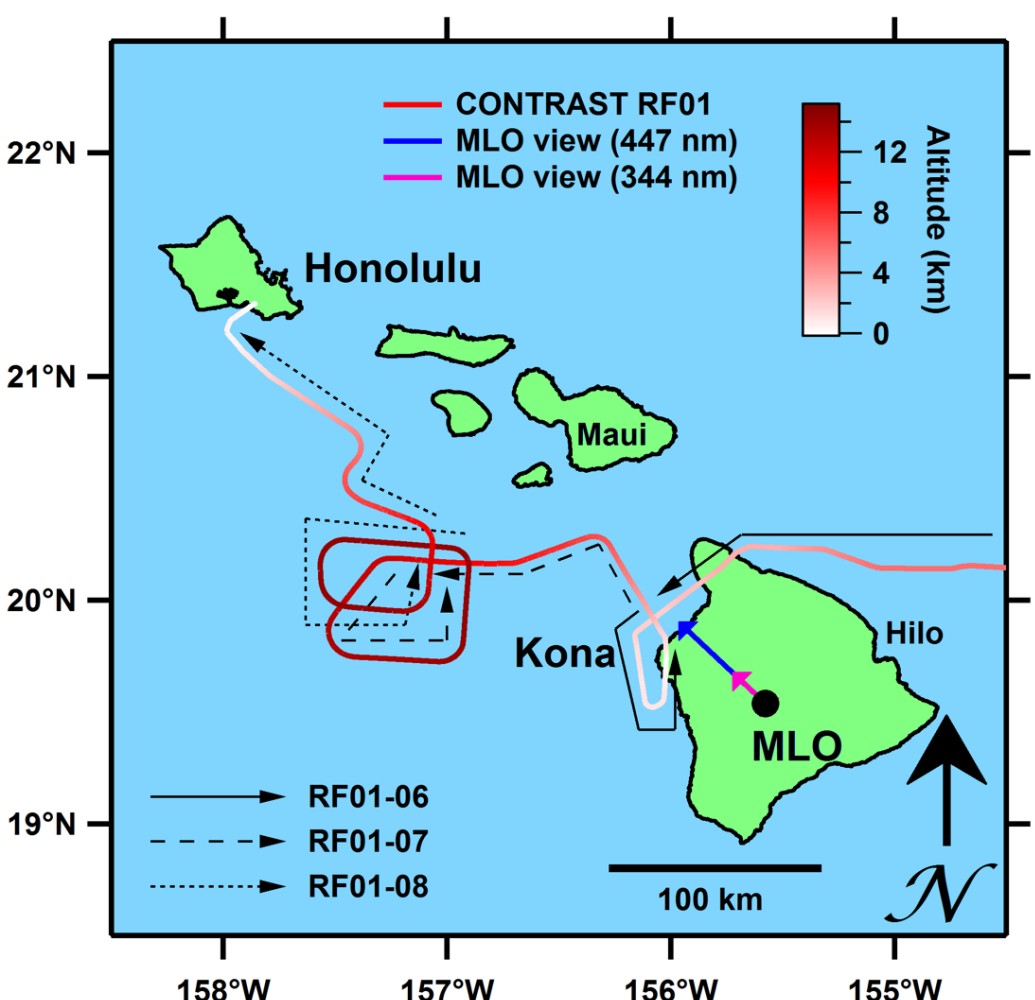

**Figure 7: Overview of CONTRAST RF01 aircraft case study in the vicinity of Mauna Loa Observatory (MLO) on January 11, 2014.**

Approaching from the east, the GV aircraft descended to a missed approach at Kona (descent is denoted RF01-06) then ascended spiraling (RF01-07) into the stratospheric intrusion (not observed on the RF01-06 to the east), finally spiraling descended (RF01-08) out of the intrusion to land in Honolulu. Color coded arrows indicate the viewing direction and approximate maximum horizontal distance sampled by the MT-DOAS near instrument altitude under typical low aerosol conditions at the wavelengths indicated. BrO observed at MLO may be transported from but is not located over the ocean.

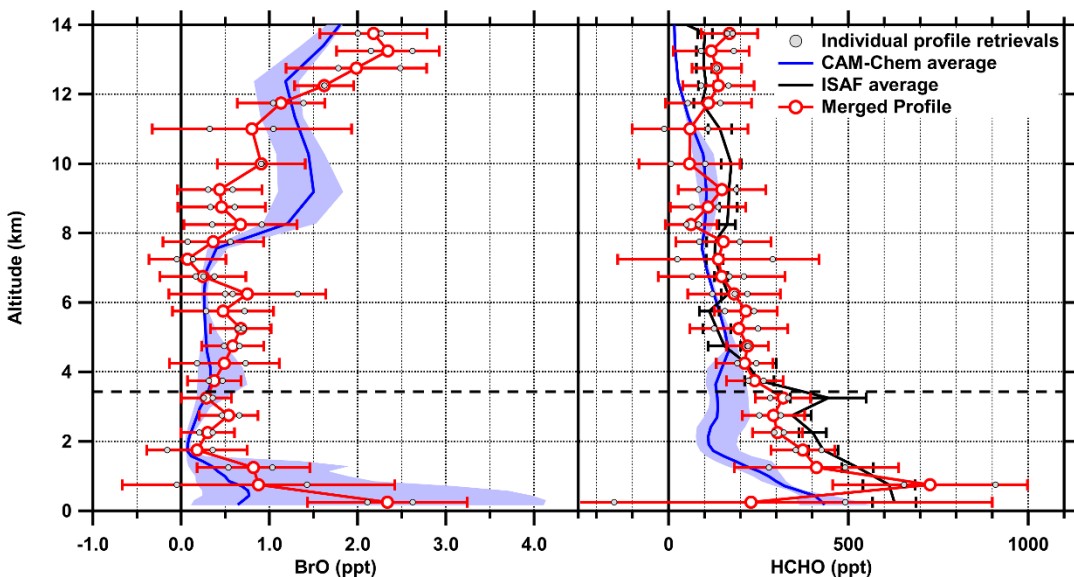

1190

**Figure 8: Retrieved BrO and HCHO columns compared with modeled profiles from CAM-Chem and measured by ISAF.**

Individual component profiles are shown as small gray points the combined profile is a weighted average. See text for details on averaging and CAM-Chem and ISAF profiles.

1195

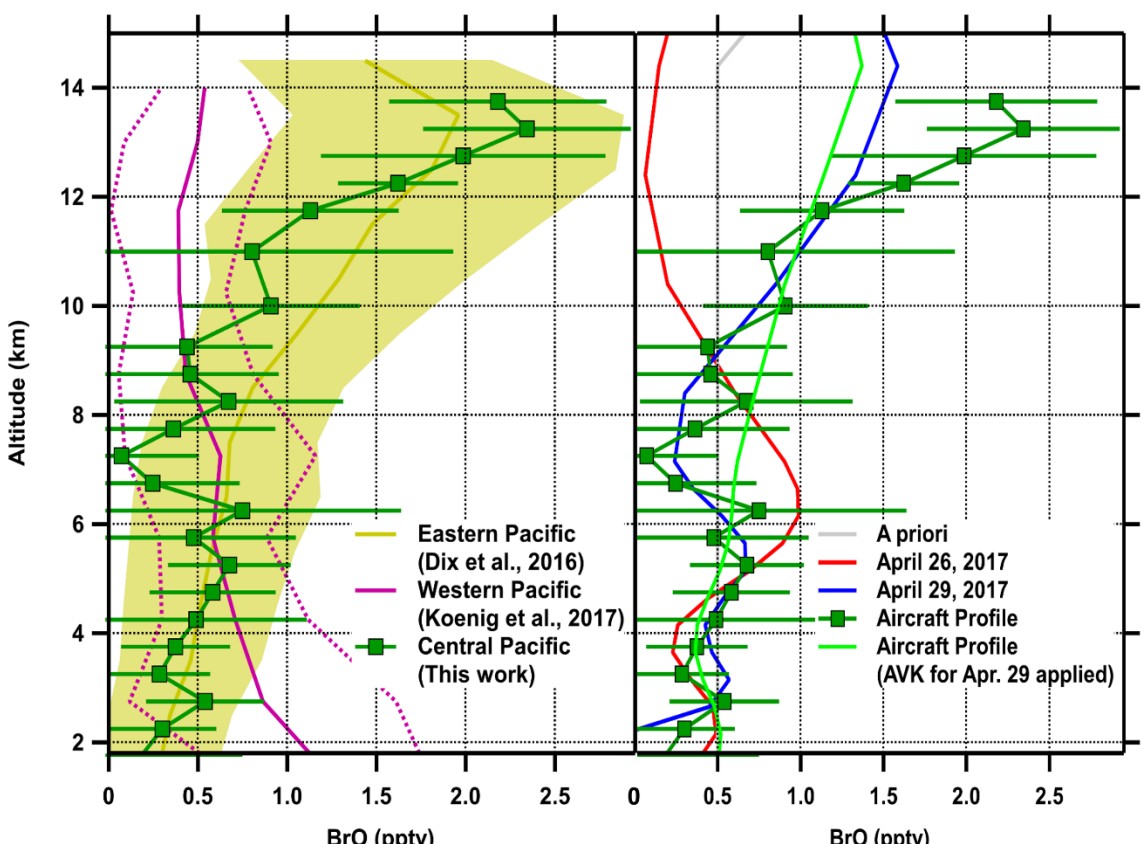

**Figure 9: Comparison of MT-DOAS retrieved tropospheric BrO profiles with AMAX-DOAS profiles.**

Left: BrO over the Central Pacific Ocean is compared with BrO profiles previously measured over the Eastern Pacific Ocean (Dix et al., 2016) and Western Pacific Ocean (Koenig et al., 2017). All profiles are measured in January and February. Right: comparison of the AMAX-DOAS and MT DOAS retrieved profiles reported here. Mountaintop profiles are shown for SZA=70°. Despite measurements occurred three years apart, and utilized different geometries, the BrO profiles for the RW-DT conditions are remarkably similar through most of the troposphere. Applying the MT-AVK to the aircraft profile resolves apparent discrepancies in the uFT.