# Peer review of "Troposphere – stratosphere integrated BrO profile retrieval over the central Pacific Ocean"

_EGUsphere, 2023_

## Author Comment (AC1)

We appreciate comments from Reviewer 1 for the improvement of the manuscript. We color code this document as follows:

**Black**: Comments from Reviewer 1.

**Blue**: Our response to Reviewer 1.

**Green**: Changes made in the manuscript reflective of Reviewer 1's comments.

**General Comments**

The manuscript entitled „Troposphere – stratosphere integrated BrO profile retrieval over the central Pacific Ocean" by Koenig et al. presents a novel retrieval algorithm for the simultaneous determination of tropospheric and stratospheric trace gases and aerosols from MAX-DOAS measurements performed on mountain tops.

Knowledge on the composition of the atmosphere, in particular in the upper troposphere and stratosphere, is crucial for the understanding of the atmospheric chemistry and its impact on climate. MAX-DOAS measurements provide a simple and cost-effective way to retrieve information on the vertical distribution of atmospheric trace gases. The mountain-top based MAX-DOAS system presented here allows for gaining enhanced information on the UT/LS region. The authors describe novel and improved retrieval methods, and the subject of the manuscript fits well into the scope of AMT.

The level of agreement of modelled and measured diurnal variability of dSCDs is impressive, indicating that the retrieval is modelling the atmospheric radiative transfer realistically. However, I find it difficult to understand the principal approach of the so-called "Time-dependent retrieval", and I feel that the respective section 2.5.3. requires substantial revision as detailed in the specific comments.

The abstract is too long and should, as an AMT contribution, focus more on the retrieval algorithm itself than on the chemistry of the UT/LS region, for which far too many details are provided.

We thank Reviewer 1 for highlighting the need for clearer explanation of the methods, especially the time-dependent retrieval also identified by Reviewer 2.

We have shortened the abstract in the revised manuscript and improved its readability. A certain length is needed to descript the key points the paper, incl. key features of the retrieval methods, figures of merit, atmospheric state changes in BrO columns and concentration profiles that are the result of time dependent optimal estimation, and the evaluation of the MT-DOAS profiles using aircraft measurements. The revised abstract is not longer than published in AMT articles elsewhere.

**Specific Comments**

L13: The term "trace element" is not appropriate for bromine since it has an impact on atmospheric chemistry not only in its elemental form.

With the reworking of the abstract this sentence and phrasing no longer occurs.

L76: It would be good if you would be more specific regarding the exploitation of the "sun motion" (i.e., increase in light path with increasing SZA).

We have reworded and further specified: …motion of the sun varying atmospheric path length and scattering attitude …

L149: It is possible to fit certain parameters or state/measurement vector elements, but it not possible to fit "degrees of freedom". Please specify what is fitted here.

We have changed degrees of freedom to width parameters

L161 and thereafter: What do you mean with the term "principal program"? Are the RTMs not just programs?

This wording was chosen so that it can be understood at the end of the paragraph that we ran both RTMs for all the data for purposes of comparison but only utilize one for the retrieval. On rereading the fact of comparison is likely understood regardless and we have edited it for clarity as follows:

Two radiative transfer codes were used for this study. For ZS-DOAS measurements, Discrete Ordinate Method Radiative Transfer (DISORT) was used and for OA measurements the Monte Carlo Atmospheric Radiative Transfer Inversion Model (McArtim) was used.

L177: Can you give an estimate how large the errors are if the atmosphere below the instrument is not considered in the DISORT model? Can this simplification be justified?

The effect is most clear in $O_4$-based aerosol retrievals which have better signal to noise. We examine this in section B of the supplement focusing on the impacts on $O_4$. The $O_4$ concentration profile is such that it presents a worst case scenario for BrO. The key effect is increased signal contribution from lower altitudes which can be partly reproduced by increasing albedo in DISORT. If DISORT were used for OA geometries, the effect could be as large as ~10% but for ZS data this agrees with McArtim results including lower altitudes to better than ~1% differences.

Section 2.3.: Please explain why you use two different RTMs in a single retrieval algorithm. This is a quite unusual approach. McArtim is capable of modelling both tropospheric and stratospheric radiative transfer also during twilight, so it is not clear what the advantage of using DISORT is.

We agree that McArtim is capable of modeling stratospheric radiative transfer at twilight, however, it presents technical challenges. We found some differences between the models even in 1D implementation around SZA = 90° (this included running McArtim with and without forcing twilight), but these could likely be resolved. The larger problem was finding a suitable 2D implementation of McArtim to use in conjunction with UVspec. In DISORT the 2D geometry is defined along the solar azimuth which is ideal for accounting for photochemical effects along the principle line of sight. The documentation of 2D and 3D implementations of McArtim is relatively sparse, but as far as we could find we are limited to lattice in altitude, latitude, and/or longitude. We successfully ran altitude-longitude, and altitude-longitude-latitude runs, which had seemingly sensible results but are much less facile to combine with the photochemical data. Furthermore, these McArtim runs were particularly computationally and time inefficient in comparison to DISORT. From the documentation available it appears that other geometries could be defined (at least in some versions of McArtim) but attempts to implement this failed. In addition, for all the McArtim versions tested we found that attempting to combine 2-D and 3-D runs with certain surface altitude and surface albedo settings had a hard requirement of supplying surface leaving radiance the required format of which we could not find documented.

The following text was added to the revised manuscript: "While McArtim is in principle capable of modeling stratospheric radiative transfer at twilight, a suitable 2D implementation of McArtim to use in conjunction with a photochemical model (i.e., UVspec) was not straightforward. We use DISORT instead since the 2D geometry is defined along the solar azimuth, which is ideal for accounting for photochemical effects along the principle line of sight."

L195: It is not clear what you mean with "layering approach".

We have added further detail. Aerosol conditions for the data in this work had significant extinction below the instrument but sub-Rayleigh extinction near and above the instrument. For these conditions, each EA was given an

altitude sensitivity mapping and the extinction profile adjusted starting from lower EA and lower altitudes. This bottom-up as opposed to top-down approach was chosen because the albedo effects from lower altitude aerosol were needed to reproduce observations at higher EA. Initial aerosol profiles were found by manually testing different boundary layer heights and AOD for a box aerosol layer below the instrument above which extinction decreased exponentially. Thereafter the sequence of $O_4$ comparisons and adjustments was run six times.

We have also added significantly more detail on the aerosol retrievals in response to Reviewer 2.

Sections 2.5.1 and 3.3.3: Is there a specific reason why SCD_ref is not simply retrieved as part of the state vector in the optimal estimation algorithm, instead of using Langley plots as an extra step?

The results in 3.3.3 hint at the reason for this choice although the reviewer is correct that other choices could have been made. When supplying $SCD_{Ref} = 2.50 \times 10^{13}$ molec. cm$^{-2}$ it recovers itself despite other deficiencies in the solution, but even more importantly it diverges away from the better solution. This suggests that the solution space has local minima and the choice and assessment of the a priori $SCD_{Ref}$ requires assessment outside a single optimal estimation in any case. The reviewer is correct that $SCD_{Ref}$ could be handled instead as an element of the state vector in principle. The relevant Jacobian elements are reasonably simple to define, but the definition of relevant a priori covariance terms is not to us obvious.

Section 2.5.3 requires substantial revision as it is not possible to understand what the actual approach is. What is the basic idea behind your approach? What is the difference between time-dependent and time-independent retrievals? The term "time-independent formulation" occurs in L291, but it is not explained anywhere what this means. What is the exact meaning of the mathematical objects in Equation 3, which of these are scalars, vectors or matrices, and what are their shapes/dimensions? Does the vector x contain profiles at a single time or are BrO profiles over a period of time which are retrieved simultaneously? What exactly is x_0 and how is it determined? Please specify in detail the individual components of the measurement vector and the state vector.

We have added a paragraph leading into and motivating the time-dependent dependent retrieval which is addressed in the response to reviewer 2. Regarding the time-independent retrieval, this is explained there also but for ease of understanding here is the integrated retrieval just outlined in Section 2.5.2. We have reworked and expanded on the definition of terms around Eq. 3 as follows.

The conventional time independent retrieval assumes constant $Br_y$ (a static atmosphere), with the only changes in BrO being those predicted by the photochemical model ($Br_y$ repartitions as a function of SZA). This assumption was ultimately found to be invalid for one day where dynamical changes in $Br_y$ were observed, in addition to chemical repartitioning (see Sect. 3.3 and 3.4 for details). We addressed this by augmenting the optimal estimation with time-dependent variables. To our knowledge such an approach has not been employed for DOAS optimal estimation before, so we describe it here in detail.

We define the time evolution of the BrO profile at time t ($\mathbf{x}$) in terms of L altitude regions (here L = 4) consisting of a weighted set of related atmospheric grid layers ($\mathbf{W^L}$) where $Br_y$ is expected to evolve consistently such that:

$$\mathbf{x} = \mathbf{x_0} \left( 1 + f(t) \sum_L \mathbf{W^L C^L} \right)$$
(3)

Where $\mathbf{x_0}$ is a vector, i.e., the BrO profile at some reference time ($t_0$) with dimensions of $1 \times 36$ for this work, and $f(t) : t \rightarrow (-1, 1]$ is a scalar time evolution function such that $f(t_0) = 0$. For convenience we choose $t_0$ to match SZA = 70° so that photochemical and dynamical effects vary on a common time axis. In principle, $f(t)$ could be indexed to the layers L and folded into the sum. However, for this work there was insufficient external information to constrain more than one choice; a single $f(t)$ function describes the relative time variation in all four altitude regions. In practice, only a linear and a ramp function form – which mirrors the stratospheric $O_3$ column on Apr. 29 were tested for $f(t)$. The choice of the codomain (-1, 1] is more generally important for reasons outlined below.

$\mathbf{W}^L \rightarrow (0, 1]$ is a matrix of altitude weights defining the mapping of altitude regions L onto the altitude grid that $\mathbf{x_0}$ is defined on constructed such that $\Sigma_L \mathbf{W}^L \leq 1$ for all altitudes with dimensions of 36×4 for this work. For this work the stricter condition that $\Sigma_L \mathbf{W}^L = 1$ for all altitudes is fulfilled. Finally, $\mathbf{C}^L$ are scaling factors describing the proportional change in the BrO profile within altitude region L, with dimensions of 1×4 for this work e.g. if an element of $\mathbf{C}^L$ has a value of 1.1, BrO increased by 10% in that layer when $f(t) = 1$.

We choose to fully constrain $f(t)$ and $\mathbf{W}^L$ as such the combination of $\mathbf{x_0}$ and $\mathbf{C}^L$ fully specifies the state vector $\mathbf{x}$. We seek to retrieve $\mathbf{C}^L$ in addition to $\mathbf{x_0}$ given a choice of $f(t)$ and $\mathbf{W}^L$.

Eq. 4: It is stated that H represents a Jacobian, but H = K_0*x is not a Jacobian Matrix but a vector in measurement space.

The language gets ahead of what is done here, we have changed the introduction text to:

We seek to derive a time-augmented Jacobian starting from the transformation $H$ in terms of a left-side transform matrix $\mathbf{K_0}$:

We have added more detail and reworked the further detailing of the time-dependent retrieval:

A close observer may notice a potential challenge posed by these equations; the weighting functions for $\mathbf{x_0}$ require knowledge of $\mathbf{C}^L$ while the weighing functions for $\mathbf{C}^L$ require knowledge of $\mathbf{x_0}$. To resolve this challenge we take advantage of the fact that the extended definition for $\mathbf{x_0}$ still contains a term which is fully independent of $\mathbf{C}^L$ and approaches the time-independent formulation as t→t$_0$. We therefore leverage the time-independent retrieval (Section 2.5.2; already photochemically indexed to t$_0$) to gain imperfect knowledge of $\mathbf{x_0}$. This retrieval already averages over the time dependence and should get us close to the true state as such we supply it as the a priori for $\mathbf{x_0}$ to compute the weighting functions. If the solution is too far from this a priori however, the computed partial derivatives might no longer be locally valid. To limit this effect we reduce the a priori covariance for the spatial variables ($\mathbf{x_0}$) by a factor of ten. In reporting results we use the spatial variables ($\mathbf{x_0}$) retrieved using the procedure in Section 2.5.2 including their corresponding AVK. The values and AVK for the time dependence vector ($\mathbf{C}^L$) are reported for the results of this second stage.

It remains to choose $f(t)$ and $\mathbf{W}^L$. Examination of Eq. 5a reveals the rationale for setting the codomain of $f(t)$ to (-1, 1], for solutions in which the entire retrieved column entirely disappears or doubles (considered reasonable bounding cases) this bounds the time dependent term to (-L, L] which for the small values of (L = 4) considered here is comparable to the time independent weight fixed at 1, hence ensuring the relative importance of measurements is at least partly preserved. In addition, by defining all $f(t)$ to have a maximum value of 1, the profile which is maximally different from that at t$_0$ is readily computed and compared. For this work we considered only linear and ramp functions, ultimately using a ramp function defined to be zero prior to 70° SZA and increasing to 1 which matches an observed trend in $O_3$ VCDs (see Supplement for details). For $\mathbf{W}^L$ logistic curves were chosen as the functional form with a logistic steepness in all cases of 2 km$^{-1}$. The atmosphere was first divided at the tropopause at 17.5 km. Then at 6 km and 10 km based on the results of retrievals using single scans and modeled behavior in CAM-chem (see Sect. 3.3 for details).

The solutions to the inversion of the time-dependent retrieval were found to be highly sensitive to the a priori supplied for $\mathbf{C}^L$ including non-physical results. We suspect that this might be because the assumptions for the staged retrieval only work where the partial derivatives are sufficiently flat, or perhaps at least smooth. This necessitated systematic sensitivity studies to find solutions which were physical as well as categorizing solutions based on minimizing any time-dependent trend in the a posteriori residuals. This methodology identified a family of solutions with similar values that met stringent criteria, from which the solution with lowest overall residual was selected (see Sect. 3.3 for details).

L317: I suppose the "high concentrations" of BrO are expected in the FT. Please specify.

We have reworded this to "increased concentrations". Prior to this study it could only be inferred that BrO should increase. What baseline concentrations are over the central Pacific was uncertain in the absence of measurements and given the variability observed elsewhere.

L354: Cross-sections do not have an optical density. Please rephrase.

We've specified that the cross-sections are "multiplied by ZS dSCDs"

L357: To what is the Aliwell fit window insensitive?

"to the choice of $O_3$ cross-section"

L363: Is a wavelength shift of 3 pm leading to any noticeable difference in the fit if the instrument has a spectral resolution of about 0.5 nm?

The difference is not statistically significant, however, it is noticeable in the third digit of fitted BrO dSCDs especially in ZS data. Even this is likely because the $O_3$ absorption is so great compared to BrO. We have included this as we believe it represents the current best practice.

L376: Please explain why the O4 scaling factor should scale with lambda^4 like Rayleigh extinction. I do not see an immediate physical reason for this.

Because $[O_4] \propto [O_2]^2$, $O_4$ signal in dSCDs overwhelmingly comes from lower altitudes typically after the final scattering event. In the single-scattering approximation the path-length from the final scattering event is inversely proportional to extinction. For low aerosol conditions extinction is well approximated by Rayleigh extinction. We have rephrased the sentence and added a reference: "This value is similar to what one might expect for a Rayleigh comparison of optical depth and pathlength of the 360 nm and 344 nm bands $(344/360)^4 = 0.83$ (Wagner et al., 2004)"

L382: Please explain what you mean with "intensity effects". Could this be instrumental non-linearity? If you suspect that NO2 is affected by such effects, then why not other trace gases, in particular if they have lower optical density?

Intensity effects is used in the context of DOAS to refer to effects which are linearly proportional to intensity rather than proportional to the ratio of intensity. Instrumental non-linearity is indeed a possible explanation, however, it is relatively unlikely as the exposure is dynamically adjusted for consistent average saturation. We've made the possibilty more explicit by specifying: Those effects could be accurately fitted as offsets from changing straylight; notably, the exposure of spectra measured at different EA is dynamically adjusted for consistent average saturation in our setup (Coburn et al., 2011).

For the UV fitting window and the measurement conditions, the differential optical density of $NO_2$ is comparably low, and we've added that it: is relevant to the small $NO_2$ signals measured in the free troposphere for which OA $NO_2$ dSCDs differ from nearby zeniths by less than three times the fit uncertainty. We suspect that the HCHO-BrO cross-talk specifically may have similar effects as in individual scans it shows similar patterns, but the "drift" effect obscures attempts to determine this more clearly.

L395ff: Here it is not clear what you mean with the terms "component" and "signal". Do they refer to the retrieved dSCDs or to the fit residuals? What exactly are replicate measurements? Do you mean subsequent measurements along the same line of sight?

We've reworded the sentence to: After optimization of the fitting window we believe that this spectral cross-talk is handled by the DOAS fit with the exception of a fast-changing anticorrelation identifiable as opposing changes in BrO and HCHO for sequential measurements of the same viewing geometry, and slow-changing opposing "drifts" in both HCHO and BrO dSCDs.

The "replicate measurements" refers to sequential repeats of EA = 0°, 30°, 90° for which the fast change is most easily identified. We've reworded to "sequential measurements of the same viewing geometry" for clarity.

L564: A "change" has no DoF.

This change does have DoF. To be more precise the time-dependent, spatially-constrained scaling factors we retrieve have information content independent of the a priori information provided which can be quantified as DoF. We took this comment as further motivation to improve the revised section 2.5.3.

L635: Here it would be good to cite Rodgers and Connor [2003].

Agreed and added

L640: In what respect is the analysis limited by the RTM calculations? In terms of accuracy? Computational time?

Specified computational time.

L643: To my knowledge, McArtim already fulfils the required capabilities listed here – see Deutschmann et al. [2011].

We agree in principle, and address this above.

Please add a "Code Availability" section stating the availability of the retrieval algorithm presented here.

Added and archived at the same locations as the data.

**Technical Corrections**

L30: near -> nearly

Changed.

L56: "BrOx adds radical species to oxidative capacity" does not make much sense. Suggestion: "BrOx increases the oxidative capacity"

Suggestion adopted

L106: I guess you mean the azimuth angle when you talk about "primary viewing direction"?

Clarified to specify azimuthal viewing direction.

L123: What do you mean with the dagger symbol as prefix for the elevation angles?

We have clarified the existing explanation at the start of the sentence which now reads:

"…where angles preceded by † are collected in the reverse azimuthal direction (+130°±2):"

L175: Stratospheric aerosol WAS modelled…

Changed

L189: Tropospheric aerosol WAS assumed…

Changed

L190: Approximated by an approximation?

Modified sentence to remove both instances of approximation as the sentence is already discussing model assumptions.

"…non-absorbing Henyey-Greenstein aerosol phase function with asymmetry parameter …"

L194: Add "The retrieval of" to the beginning of the sentence

Changed

L393: The term "method-based anticorrelation" is not clear to me.

We have rephrased the sentence for further clarity:

This similarity in measurement leads to an empirical anticorrelation, the characterization of which is further confounded by chemical coupling of BrO and HCHO via reactions of Br atom with HCHO and other aldehydes, which often correlate with the latter, suppressing BrO formation and creating a chemical anticorrelation."

L404: The part of the sentence after the semicolon is without any context.

We have replaced the semicolon with a period and combined the following clause with the following sentence for which it provides context.

L409: I suggest to add that the additional HCHO absorption feature is at 330 nm.

We have specified that the feature is "peaking between 329 nm and 330 nm" to ensure it is understood that windows starting at 330.0 nm do not capture the peak.

L497: comparison -> difference

Changed

L506: "While it is clear the retrieval … can be further improved" is grammatically incorrect. Is a "that" missing?

We have add "that" for clarity.

L585: Explain abbreviation "KOA".

Changed to "Kona airport (KOA)"

We wish to note that when conducting sensitivity studies in response to comments from Reviewer 2, we discovered that the DoF reported for April 29 had been erroneously entered for different retrieval settings than those used. We reviewed all other results and confirmed that this error was limited to only the DoF. The reviewer may wish to check the revised numbers.

**References**

1. Deutschmann et al., "The Monte Carlo atmospheric radiative transfer model McArtim: Introduction and validation of Jacobians and 3D features," J. Quant. Spec. Rad. Trans., vol. 112, pp. 1119–1137, 2011, doi: 10.1016/j.jqsrt.2010.12.009.

2. D. Rodgers and B. J. Connor, "Intercomparison of remote sounding instruments," J. Geophys. Res, vol. 108, no. D3, pp. 4116–4229, 2003, doi: 10.1029/2002JD002299.

---

## Author Comment (AC2)

We appreciate comments from Reviewer 2 for the improvement of the manuscript. We color code this document as follows:

**Black**: Comments from Reviewer 2.

**Blue**: Our response to Reviewer 2.

**Green**: Changes made in the manuscript reflective of Reviewer 2's comments.

"Troposphere – stratosphere integrated BrO profile retrieval over the central Pacific Ocean" by Koenig et al. utilizes a mountaintop DOAS to retrieve vertical profiles of BrO throughout the troposphere and stratosphere in the Central Pacific. This method utilizes an elevated MAX-DOAS instrument to increase sensitivity to large portions of the atmosphere and builds on previous BrO observations in the Western and Eastern Pacific. This manuscript also develops a novel approach to BrO differential slant column density retrievals to increase the stability of their observations. The paper also discusses the development of a time-sensitive vertical profile retrieval. These methods help to better understand the amount of BrO in the background free tropophsere and other often unobserved areas of the atmosphere with utility for anthropogenic pollution monitoring in large cities.

Ultimately, this is an important work that would be beneficial to extend to other areas. However, some specific information on the retrieval methods is under-described or omitted. Similarly, the mathematics of the time-dependent retrieval are not well defined. The method described here would be difficult to reproduce based on the current state of this paper. The "Results and Discussion" section also skews too much toward discussion and omits some important results, which should be more of the focus in this work.

We thank Reviewer 2 for highlighting the need for clearer explanation of the methods, especially the time-dependent retrieval also identified by Reviewer 1. The need for more complete description of the results is also useful. We have edited Sections 2.5.2 and 2.5.3 in the revised manuscript with an eye towards clarity for reproduction of our work by others (method section), and a highlighting of the benefits of the developed methods (results section).

**Major Comments:**

The BrO fit routine seems somewhat arbitrary. I understand using a fit that results in less negative BrO values, however it is also important to discuss how the fit uncertainty is affected by these decisions. Similarly, it is not mentioned if the negative values retrieved for some specific elevations are outside of the retrieval uncertainty. If not, then the BrO fit constraints seem like they may not be necessary. I am also somewhat concerned that two of the fit constraints depend on radiative transfer modelling as this is a source of uncertainty. Similarly, I am wary of constraining three parameters in the BrO fit while also using a quite high polynomial order. Ideally, a sensitivity study would be conducted on how the three constrained fit parameters impact the retrieved BrO dSCD. I understand that this is likely not feasible on your timeline, so I would at least ask for detection limits/uncertainties for the different trace gases (BrO, $O_4$, HCHO, $NO_2$). This also leads to another issue. Utilizing just the BrO fit uncertainty for your measurement error covariance matrix likely underestimates the true uncertainty of these retrievals. As I have discussed, there is likely more uncertainty from these sources that is unaccounted for, but the uncertainty of the BrO cross-section should also be considered. Again, I understand it is not possible to update your profile inversions on this time scale, but a more detailed discussion of error propagation would be appreciated.

As it happens we conducted the requested fit sensitivity studies in the course of developing the methods. The overall effect of the fit constraints is captured in Fig. S5, however, we appreciate that further detail and discussion of the individual constraints is useful for the reader to understand the magnitude of the effects involved.

We have added the following detail to a new section in in the supplement "**Sensitivity studies assessing the uncertainty of fit constraints on BrO dSCDs**" regarding the $O_4$ constraint:

The $O_4$ constraint changes BrO dSCDs by an average and standard deviation of $+0.7\pm4.4\times10^{12}$ molec. cm$^{-2}$, less than the average fit uncertainty of $6.1\times10^{12}$ molec. cm$^{-2}$ and smaller on an individual basis in almost all instances. For ZS fits, the effect is somewhat larger ($-1.1\pm5.4\times10^{12}$ molec. cm$^{-2}$) but proportionally still smaller than the average fit uncertainty ($10.0\times10^{12}$ molec. cm$^{-2}$). Given that the effect is small, and the constraint employed emerges empirically from unconstrained $O_4$ fits, the use of the RTM to estimate the constraint does not add significantly to the uncertainty, which is dominated by the fit errors. The key impact of the $O_4$ constraint is to give more consistent fits based on the complete knowledge available about $O_4$, which are critical to further constraints discussed below.

The following regarding the $NO_2$ constraint in the maintext:

"…for which MAX $NO_2$ dSCDs differ from nearby zeniths by less than three times the fit uncertainty"

And in the supplement:

On average, the $NO_2$ constraint counteracts the $O_4$ constraint but with less variability ($-0.7\pm2.2\times10^{12}$ molec. cm$^{-2}$ effect on BrO dSCDs). The changes while small systematically affect specific EAs with EA≤0° mostly increased by $1-2\times10^{12}$ molec. cm$^{-2}$ BrO and EA=30° typically decreased by a similar amount. Hence, while less than fit uncertainty in almost all instances, the $NO_2$ constraint can ultimately impact the BrO profile retrieval, and is implemented here in a radiatively consistent way leveraging the best knowledge available about $NO_2$.

More detail on HCHO and the overall constraint is provided in the following expanded paragraph:

"The effect of constraining HCHO is much greater than those of $NO_2$ and $O_4$ which roughly counteract each other (on average) in their effect on BrO dSCDs; HCHO sensitivities dominate the pattern observed in Fig. S5. The overall effect of constraining HCHO is almost always to increase BrO dSCDs (Fig. S5 data are above the 1:1 line) which given the low SZA zenith reference spectrum is easier to understand physically. The effect of the HCHO constraint – including fast and "drift" components – is $1.5\pm0.9\times10^{13}$ molec. cm$^{-2}$, dominating the overall effect of the constraints which have the same mean and standard deviation. Unlike the other constraints, this is on average almost 2.5 times the fit uncertainty of BrO and dominated by the "drift". Secondary effects are most apparent for larger EA (≥12°) for which dSCDs are "pulled" as a result of the constraints to non-negative, physically meaningful, values relative to the BrO dSCDs of the nearest zenith spectrum. In all instances, these smaller changes are well within the range of uncertainty. They are, however, important to the BrO profile retrieval as they act on relatively small separations between these EA."

With additional detail in the supplement:

The top panels of Fig. S6 illustrate the importance of the HCHO constraint. Based on the a posteriori HCHO SCDs, the impact of the "drift" in the HCHO-BrO spectral cross-talk varies to as much as $\sim5\times10^{15}$ molec. cm$^{-2}$ HCHO at SZA = 70° on Apr 26 and is relatively constant around $\sim1\times10^{16}$ molec. cm$^{-2}$ HCHO on Apr. 29. Applying the constraint of HCHO lowers the HCHO dSCDs by these corresponding amounts and consequently increases BrO. The magnitude of the cross-talk varies, but for this drift BrO dSCDs roughly increase by $2\times10^{-3}$ of the corresponding HCHO dSCD decrease, i.e. the $\sim1\times10^{16}$ molec. cm$^{-2}$ HCHO decrease on Apr. 29 corresponds to $\sim2\times10^{13}$ molec. cm$^{-2}$ BrO increase.

Regarding the information content of the DOAS fit more generally we now note in the text that: "The wider window necessitates an order 7 polynomial due to its width and the Huggin's band absorption gradient but at ~45 times the FWHM of the slit function has more than sufficient information for fitting the 17 linear absorption parameters."

Reviewer 2 is correct that systematically addressing the effect of different measurement covariance matrices on the retrieval – especially if employing fully all steps of the retrieval – would be very time consuming. In particular, searching the stability of the time-dependent a priori space for changes is not practical. In brief, we believe the BrO SCDs are correctly chosen as already thoroughly justified but that modifying the measurement covariance is a useful (and practical) sensitivity study. "Sensitivity studies were conducted accounting additionally for the uncertainty in the BrO cross-section (~10.5%) and the constrained DOAS fits."

We've added a new supplemental figure S8, and added the following paragraph in Sect. 3.3.4:

Accounting for additional sources of uncertainty in the measurement covariance ($S_\varepsilon$) highlights the robustness of the retrieval. Accounting for the uncertainty in the BrO absorption cross-section (~10.5%; Fleischmann et al., 2004) as well as the change in BrO dSCD from the $O_4$ and $NO_2$ constraints and the non-drift component of the HCHO fit constraint (order of $10^{12}$ molec. cm$^{-2}$) fundamentally lowers the signal to noise, leading the retrieval to generally trend slightly toward the a priori and be smoothed over (Fig. S8). The "drift" observed in the HCHO-BrO cross-talk is clearly an instrumental effect, however, we account for it in a further sensitivity study to further probe the most robustness of the retrieval. The effect on Apr 26 is, as expected, minimal and while more than one DoF is lost on Apr 29, the change in VCD is small and the increased BrO in the upper free troposphere is still retrieved. Even accounting extremely conservatively for uncertainty, the retrieval already improves on previous BrO retrievals.

With additional detail in the supplement:

We next examine the effect of incorporating these constraints as part of an expanded measurement covariance in the time-independent BrO profile retrieval as well as the BrO cross-section uncertainty. Applying the fit constraints leverages observed information (from other fitting windows and inversions) and cross-section uncertainty are systematic rather than random, as such we do not use them in the default retrieval. We first consider the BrO cross-section uncertainty and the $O_4$ and $NO_2$ constraints. The major features of the retrieved profiles are retained and the changes in BrO VCD are minimal: (-0.2 and -1.0)×$10^{12}$ molec. cm$^{-2}$ on Apr 26 and 29 respectively. The larger effect on Apr 29 is mostly the result of smoothing the profile shape at a lower concentration below ~7.4 km. The reduced signal to noise is reflected as a decrease in DoF by 0.61 on Apr 26 and 0.56 on Apr 29 with each day retaining 4.99 and 5.00 DoF respectively. The loss of DoF is minimal near instrument altitude and is greatest in the uFT and stratosphere. Next we further account for the much larger effect of the HCHO constraint. As expected the effect on Apr 26 is small, with the profile smoothing toward the a priori, a decrease to 4.77 DoF, and a change in BrO VCD of <0.1×$10^{12}$ molec. cm$^{-2}$ compared to the default retrieval. Even on Apr 29 when the effect of the "drift" is greater than many individual dSCDs, 3.87 DoF are still retrieved with almost all the loss from the troposphere where the change in signal to noise is greater. Nonetheless, the change in BrO VCD is still small (-0.9×$10^{12}$ cm$^{-2}$) and the elevated BrO in the upper troposphere is still retrieved.

We stand by the current construction of the covariance as best since it ultimately is leveraging measured information, albeit indirectly. The moderate sensitivity study clear to the reader what the effects of the constraints are while the very conservative case is included more for completeness and transparency.

We wish to note that when conducting these sensitivity studies we discovered that the DoF reported for April 29 had been erroneously entered for different retrieval settings than those used. We reviewed all other results and confirmed that this error was limited to only the DoF. The reviewer may wish to check the revised numbers.

It is unclear how the aerosol profiles are retrieved for the MT-DOAS observations. It is stated that the difference between measured and modelled $O_4$ was calculated for each scan. However, it is not clear how this is used to retrieve a particle extinction profile? Is an inversion done for each scan?

We did not conduct an inversion by optimal estimation, in part because $O_4$ measurements in the limb geometry generally agree within few percent with those calculated in a Rayleigh atmosphere. There is very little aerosol above the site. This is by design, and a key advantage of the MT-DOAS geometry.

In response to the reviewer comment, we have revised text in Section 2.3.2. to clarify this fact, the level of agreement, and the methods used. We also have renamed Section 3.3.1. "Aerosol Profile Retrieval" to eliminate possible confusion. The revised manuscript now makes it clear that aerosols/clouds/terrain below the instrument altitude can feed back into the $O_4$ observed in upward pointing EAs, resulting in effects on the order of 10% higher $O_4$ that is measured than modeled. Such small differences are difficult to even measure outside fit error at 360nm (typically on the order of 5-7%), but are observable at 477nm (2-3% fit error) due to the reduced Rayleigh scattering, and thus larger contrast to detect aerosols at these longer wavelength. At the relevant shorter wavelengths of the BrO retrieval stronger Rayleigh scattering is partially masking possible effects from aerosols, which are regardless treated here in a radiatively consistent way using the information at longer wavelengths.

I am still unsure how the time-dependent retrieval works. You explain how the Jacobians are determined, but outside of that I am not sure how the retrieval works. I also do not know what the inversion retrieves. It seems that the inversion retrieves $\mathbf{x_0}$ and $\mathbf{C^L}$, but I'm not sure how the inversion retrieves both values. Are two inversions performed, or is only one performed to retrieve both parameters? If so, it seems that the inversion could settle on a local minimum rather than the "true" solution. How is this accounted for? I also do not quite understand what this retrieval tells you, and how you combine the parameters to retrieve a profile. You also state that the retrieval of $\mathbf{C^L}$ is particularly sensitive to the apriori profile. Just how sensitive is it, and how confident are you in the apriori value used? I also do not understand the physical representation of $\mathbf{C^L}$, and the choice of apriori described in the supplement seems somewhat arbitrary. It seems that this sensitivity could impact the retrieval uncertainty in a way not accounted for in the output covariance matrices. I also do not fully understand the ramp function. You state that the domain is from -1 to 1, though it seems in practice that it is actually from 0 to 1. I also do not understand how this is tied to the $O_3$ VCD, particularly since this retrieval is not described. Line 80 also says that this method accounts for non-photochemical diurnal variability. That was not the impression that I got from this method. Lastly, it seems that the utility of this method is that it utilizes the change in zenith retrievals as a function of time, whereas fitting each scan with a local zenith would remove sensitivity to the stratosphere. If this is the case, I feel this method could be motivated a little better.

We appreciate this comment. The revised manuscript now gives a more detailed description of the time-dependent retrieval. Please also see our response to reviewer #1 on this point. In particular, we begin by better motivating the approach by comparison to retrievals of individual scans in Sect. 2.5.2:

The conventional approach to retrieving trace-gas profiles changing in time from data acquired by a MAX-DOAS instrument is to retrieve separate profiles from individual OA angle scans – often in a moving reference analysis to minimize the effect of signals mostly captured by ZS data. Where information content is limited, scans might also be averaged or otherwise combined. One limitation of such an approach is how to combine it with a stratospheric profile retrieved using ZS-DOAS data. Using the nearest zenith reference removes most dependence on the stratospheric profile but to our knowledge existing approaches effectively impose a time-dependence on the stratosphere (most typically constancy) as does our time-independent retrieval. Furthermore, the constraint although conceptually on the stratosphere also includes altitudes where the information content from an individual OA scan is limited and ZS variation contributes significantly such as the upper troposphere and the various constraints are tied together. The different scans are contingent on the imposed trend and not statistically fully independent which can obscure the statistical significance of comparing scans. Here we define an alternate approach where a consistent time function $f(t)$ for changes in the profile is used but different layers in the atmosphere can vary separately.

We address specific changes to language around the equations in responses to reviewer 1 who had similar comments which were more specifically posed in that portion of the section.

**Minor Comments:**

Line 43: The C-shaped Western Pacific BrO profile is only mentioned here and in the conclusion. Considering you compare and contrast with Eastern and Western Pacific profiles later, this needs to be introduced in the introduction and cited here.

We've added a sentence to the introduction: Over the Pacific, previous tropospheric measurements have found that BrO mixing ratios increase roughly linearly with altitude over the eastern Pacific being near or below detection in the boundary layer and greatest below the tropopause (Volkamer et al., 2015; Wang et al., 2015; Dix et al., 2016), while measurements by DOAS (Koenig et al., 2017) and other methods (Le Breton et al., 2017; Chen et al., 2016) find a more C-shaped profile over the western Pacific.

Line 74: Define MT-DOAS here.

Changed

Line 123-124: I do not understand this sentence.

We have rephrase for clarity: Spectra in both directions were analyzed, however, it was subsequently found that the two viewing directions could not be reproduced simultaneously with 1D radiative transfer modelling and the data from the reverse direction are not reported here.

Line 126: Local time would be preferable. This is later in the morning than what data?

We have added the local time in addition to UTC (while Hawaii does not observe daylight saving time we prefer UTC for clarity). We've clarified that this the reference is "is later in the morning than the data presented in this work".

Line 127-128: What are the moving reference analysis and fixed reference analysis?

We've clarified: For moving reference analyses, the fixed reference analyses are adjusted by the fitted zenith spectra linearly interpolated in time which was found to obtain results not statistically distinguishable from irradiance interpolation but is much more time efficient.

Line 141: What is the naming convention of the flight segments? Later you use RF01-06 for example. It would be nice to introduce this here.

We add some brief detail here: Flight segments are designated following a system described more fully in Koenig et al., (2017), in brief monotonic ascents and descents for a given flight are assigned sequentially as (RF##-aa) such that all ascents have odd numbers for aa and all descents have even-numbered aa.

Line 142: Local time would be preferable.

Added

Line 156: Regarding "full non-linear treatment," are there limits applied to the shift and squeeze?

There are not strict limits applied as this is not easily implemented in QDOAS. Results for the unconstrained fits were compared with WinDOAS fits with previously determined constraints following Coburn et al., (2011) and found to be consistent, the shift and squeeze were subsequently monitored for any odd behavior.

Line 168: Why do you specify these SZAs and altitudes?

For stratospheric altitudes at greater SZA especially, we started to observe larger deviations between the models. For lower altitudes the radiative effects including albedo and ground altitude become relevant. We provide more detail regarding the comparison in response to Reviewer 1.

Line 261-264: It seems like you set the apriori to 50% of the Theys et al., 2009 climatology with a 50% uncertainty above 17.4 km. This would indicate that the climatology value at these altitudes is not within the apriori uncertainty. What was the reason for this choice?

This point is well taken. We did test using the climatology and 100% uncertainty in the stratosphere during preliminary investigation. These settings were found to be more stable when varying other a priori parameters. We add the context that: This results in the diagonal elements of $S_a$ being relatively similar with altitude compared to using the climatology.

For an absorber present in both the stratosphere and troposphere but at highly variable concentrations, signal will be attributed to altitudes where the a priori is more accommodating of change. Therefore, a relatively "flat" a priori uncertainty is preferable all else being equal.

Line 487: Reference the figure at the end of this line.

Changed.

Line 510: Is this technically 3.2 DOF? I think I may prefer seeing the total DOF of the entire profile here before you break it down into different altitude layers.

We have chosen to instead introduce Table 1 at the start of the paragraph. This includes both the total and component DoF for ready reference.

Line 561: is 7e13 molecules/cm$^2$ correct? I am not sure what this value is, but it is much larger than any other value here.

This was an order of magnitude error, the correct value is 7e12, now reported as $0.70\pm0.14)\times10^{13}$ for consistency.

**Technical Comments:**

I often think of MAX-DOAS and ZS-DOAS as the instrument itself. I would suggest changing references to the different observations to zenith and off-axis or something similar for clarity (e.g. lines 161 and 187).

In the case of MAX-DOAS we agree that off-axis (which we abbreviate OA) is best distinguished from the instrument and have changed all instances accordingly. We've retained zenith sky, as there are no ZS-DOAS instruments mentioned outside the introduction and the use of two abbreviated terms makes it easier for the reader to identify and understand when results are compared and contrasted, aiding comprehension despite any risk of confusion.

RW-DT and RWB seem to be used interchangeably. If this is the case, the text should commit to one name. If this is not the case, the difference needs to be clarified.

Most instances of RWB are indeed interchangeable with RW-DT, and we have changed these accordingly. In the limited instances where RWB is not interchangeable we have defaulted to spelling out "Rossby wave breaking".

Discussion of results is often confusing as a range of values is often given (e.g., line 31, 554, 555, 556, etc.). Are these the ranges from the different scans? If so, why are they not always lowest-largest? These values need a little more context.

We believe the misunderstanding here originates from a lack of clarity about the meaning of the retrieved time-dependence variables. We have modified the explanation of this extensively elsewhere. We have sought to draw a distinction between use of n-dashes to communicate ranges and the word "to" to communicate a time-vector for change. We leave ranges unedited but clarify the pattern employed for the latter case as follows: "…whether examined at SZA = 70° or at the minimum SZA: base case (SZA = 70° to SZA = min.;2.2 to 2.3 ± 0.2)×10$^{13}$ molec. cm$^{-2}$ …"

Line 305: I am curious how you reduce the apriori uncertainty by a factor of 10 and result in a higher retrieval DOF. Can you explain this?

We address this in the revised methods section on the time dependent retrieval (2.5.3). In brief the spatial DoF are used from the first time-independent retrieval when the uncertainty was still a factor of 10 greater, the new DoF are specifically for the time-dependence variables.

Section 3.1 does not really contain any results. It may be more appropriate in Section 2.

We have kept this subsection in the results because it includes information on the identification of the case studies which requires introduction of measured results to address, particularly the measured $O_3$ VCDs and consultation of modeled meteorology.

Was the impact of $O_3$ reference temperature considered in Section 3.2?

We have clarified that: We employed the same treatment for temperature dependence and orthogonalization (Table S1) for all cross-sections to focus on the cross-sections themselves. We have not systematically probed using different temperatures outside of 223K and 243K for $O_3$ cross-sections as we believe the former is sufficiently similar to $O_3$-profile weighted temperature and the latter captures most of the variability without loss of linearity.

I am still unclear on why AMFs and SCDs were used to constrain HCHO in the BrO fit.

From the existing text we wish to clarify "The SCD of the fixed reference was then subtracted to yield HCHO dSCDs which were used to constrain the BrO fit." The effects of this constraint and therefore its importance were briefly presented in the second paragraph of Sect. 3.3.2 and we have reworked and expanded this paragraph to provide further detail:

The effect of constraining HCHO is always much greater than that $NO_2$ and typically more impactful than constraining $O_4$, and dominates the pattern observed in Fig. S5. The overall effect is almost always to increase BrO dSCDs (Fig. S5 data are above the 1:1 line) which given the low SZA zenith reference spectrum is easier to understand physically. The top panels of Fig. S6 illustrate the importance of the HCHO constraint. Based on the a posteriori HCHO SCDs, the impact of the "drift" in the HCHO-BrO spectral cross-talk varies to as much as $\sim5\times10^{15}$ molec. cm$^{-2}$ HCHO at SZA = 70° on Apr 26 and is relatively constant around $\sim1\times10^{16}$ molec. cm$^{-2}$ HCHO on Apr. 29. Applying the constraint of HCHO lowers the HCHO dSCDs by these corresponding amounts and consequently increases BrO. The magnitude of the cross-talk varies, but for this drift BrO dSCDs roughly increase by $2\times10^{-3}$ of the corresponding HCHO decrease, i.e. the $\sim1\times10^{16}$ molec. cm$^{-2}$ HCHO decrease on Apr. 29 corresponds to $\sim2\times10^{13}$ molec. cm$^{-2}$ BrO increase. This is similar to the individual fit uncertainty, but significant as a systematic effect. Secondary effects are most apparent for larger EA ($\geq12°$) for which dSCDs are pulled to non-negative, physically meaningful, values relative to the nearest zenith spectrum. In all instances, this is well within the range of uncertainty, nonetheless, this change is important to BrO as it acts on relatively small separations between these EA.

Section 3.3.1 – Low on results. I would prefer discussion of the retrieved profiles including the DOF.

Please see our response to the general comment, the retrieval is not an inversion and does not have DoF as such.

Section 3.3.3 – Is the SCD$_{ref}$ used to add to the dSCDs to retrieve the BrO profiles with SCDs? If so, this is not entirely clear. Also, the fit routine described here is not entirely clear. It seems like the SCD$_{ref}$ is used to calculate the AMFs, where the resulting Langley plot is used to determine a SCD$_{ref}$. However, the AMF calculation is more dependent on profile shape and the distribution of BrO in the troposphere compared to the stratosphere. Based on the plot, the choice of 2e13 molecules/cm$^2$ seems to be the best choice, but the fact that the SCD$_{ref}$ input results in the same SCD$_{ref}$ output is not as significant without knowing how the other parameters are constrained.

To further clarify regarding the use of SCD$_{Ref}$ we have added at the end of Sect. 2.5.1: "SCD$_{Ref}$ is added to the dSCDs to obtain SCDs for optimal estimation." We also reiterate at the opening of Sect. 3.3.3: "To obtain SCDs for optimal estimation requires determination of SCD$_{Ref}$."

We do not agree with the Reviewer's contention that: "the AMF calculation is more dependent on profile shape and the distribution of BrO in the troposphere compared to the stratosphere." For MAX-DOAS data collected for SZA < 70° the photochemical correction is minimal and typically negligible. As such while we did not run additional chemical modeling we did conduct sensitivity studies with different tropospheric profile shapes. This included using the final retrieved profile for the base case (Apr. 26 from which the reference is taken) to assess consistency. We add relevant detail in the following sections.

We add in Sect. 3.3.3.: "Substituting the final retrieved profile shape for the climatological BrO profile retrieves identical results to within the precision of the significant figures."

Regarding the troposphere and other inputs to the photochemical Langley plot, we agree that it can be difficult to follow the internal references from Sect. 3.3.3 to 2.5.1, and in turn to 2.3.1 to find the details on profile shape sought by the reviewer. To quote from there regarding the troposphere: "PSCBOX was run with 20 altitude levels between ~10 and ~55 km (altitudes below 10 km were assumed to have the same chemical partitioning as the lowest level) with chemical species from the SLIMCAT 3-D chemical transport model (Chipperfield, 2006; Hendrick et al., 2007). The model has been updated to reflect the latest bromine chemistry taken from the JPL 2015 compilation (Burkholder et al., 2015)." To aid in this we have retitled Sect. 2.3.1 to "**DISORT with PSCBOX**" in hopes of helping readers find this desired detail. We have also added in Sect. 2.5.1: "Particular instances of DISORT PSCBOX are selected based on the month of year, latitude, and bromine loading and interpolated to more precisely match a preliminarily chosen value of VCD(70°)."

Line 554-571: These results are difficult to interpret. It is difficult to tell what the ranges mean, and I believe you alter between giving values and changes in values. The section needs to be streamlined to be clearer. For example, I am not sure what line 561-564 is meant to indicate?

We appreciate that this section can be difficult to follow. We've sought to retain specific quoted numbers for values and changes as we believe these are potentially useful as key findings to readers. We have made numerous smaller changes to the text to aid in communication, breaking up larger sentences and restructuring for better clarity. The revised passage is now as follows:

RW-DT events involve the movements of mid-latitude air toward the tropics which might be expected to increase the BrO VCD, however, the difference in BrO VCD is not significant whether examined at SZA = 70° or at the minimum SZA: base case (SZA = 70° to SZA = min.; $2.2$ to $2.3 \pm 0.2) \times 10^{13}$ molec. cm$^{-2}$ vs RW-DT ($2.6$ to $2.4 \pm 0.3) \times 10^{13}$ molec. cm$^{-2}$. The stratospheric BrO VCD differs by $5 - 10\%$ between the two days, ($1.46$ to $1.47 \pm 0.08) \times 10^{13}$ molec. cm$^{-2}$ in the base case, and ($1.61$ to $1.55 \pm 0.08) \times 10^{13}$ molec. cm$^{-2}$ for the RW-DT (Fig. 6), broadly consistent with the 7% difference in the $O_3$ VCDs for SZA < 70° (Fig. S8). Given the observed increase in $O_3$ VCDs during April 29, it could be expected that the observed decrease in stratospheric BrO VCDs over the morning of April 29 must be compensated by a tropospheric increase resulting from the RW-DT, however BrO SCDs are actually lower than expected by the time-independent retrieval (Fig. 4). Consistent with this the time-dependent tropospheric BrO VCD decreases from ($1.01 \pm 0.14) \times 10^{13}$ molec. cm$^{-2}$ to ($0.85 \pm 0.17) \times 10^{13}$ molec. cm$^{-2}$. Counterintuitively, given the expectation that RW-DT is typicaly conceived to inject BrO into the troposphere, BrO decreases by ($0.15 \pm 0.17) \times 10^{13}$ molec. cm$^{-2}$ from ($1.01 \pm 0.14) \times 10^{13}$ molec. cm$^{-2}$ in the RW-DT case mostly in the mid to lower-FT (>80% of the change), the altitudes furthest from the stratosphere. That tropospheric BrO increases slightly, ($0.13 \pm 0.16) \times 10^{13}$ molec. cm$^{-2}$ over ($7.00 \pm 0.14) \times 10^{13}$ in the base case demonstrates the capacity of the retrieval to produce such an increase when reflective of the underlying data. The small to negligible change in the stratosphere and upper FT for the RW-DT has 0.86 total DoF. This suggests that while the $O_3$ VCD provides a reasonable estimate of the change in stratospheric BrO VCDs between the two days, it does not readily predict changes during the RW-DT event.

Line 592: The plot seems to indicate that there is low measurement sensitivity for the retrieval in the lowest layer of the atmosphere. That would explain the large uncertainty.

We believe that the Reviewer is referring to Fig. 8, consulting Fig. S14 in the supplement shows that measurement sensitivity is nominally the same as for other low altitudes near the surface. The source of the uncertainty is observed variability in the underlying HCHO dSCDs for the RF01-07. An expert reader examining the HCHO dSCDs in Fig. S12 might identify that the sequence of the first four dSCDs during the rapid ascent following the missed approach likely play a key role. Given the small number of points of most importance and not able to rule out that these reflect horizontal rather than vertical variability we believe the existing statement that "the large propagated uncertainty in the optimal estimation indicates limitations of the retrieval also likely play a role."

Line 628: I'm unclear why you would apply the MT-DOAS AVKs to the aircraft profile. Are you indicating this is how the MT-DOAS would view this profile?

Yes, this is the intention. We have added a sentence when Fig. 9 is introduced to make this fully transparent: "We apply the AVK from the MT-DOAS retrieval to the AMAX-DOAS profile to better compare how the two profiles would appear on the same instrument."

Line 632: Which model predicted a more intense RWB?

We used CAM-Chem data as outlined in the figure caption and methods. We make this explicit here for clarity.